# Phenotypic bistability in *Escherichia coli*'s central carbon metabolism

Oliver Kotte[1,†], Benjamin Volkmer[1,†], Jakub L Radzikowski[2] & Matthias Heinemann[1,2,*]

## Abstract

Fluctuations in intracellular molecule abundance can lead to distinct, coexisting phenotypes in isogenic populations. Although metabolism continuously adapts to unpredictable environmental changes, and although bistability was found in certain substrate-uptake pathways, central carbon metabolism is thought to operate deterministically. Here, we combine experiment and theory to demonstrate that a clonal *Escherichia coli* population splits into two stochastically generated phenotypic subpopulations after glucose-gluconeogenic substrate shifts. Most cells refrain from growth, entering a dormant persister state that manifests as a lag phase in the population growth curve. The subpopulation-generating mechanism resides at the metabolic core, overarches the metabolic and transcriptional networks, and only allows the growth of cells initially achieving sufficiently high gluconeogenic flux. Thus, central metabolism does not ensure the gluconeogenic growth of individual cells, but uses a population-level adaptation resulting in responsive diversification upon nutrient changes.

**Keywords** bistability; flux sensing; metabolism; noise; persisters
**Subject Categories** Metabolism; Quantitative Biology & Dynamical Systems
**Mol Syst Biol. (2014) 10: 736**

## Introduction

Since the early studies of bacterial physiology, inoculation of a bacterial population into a new medium has been known to result in a period lacking apparent growth prior to growth on the new carbon source (Monod, 1949). This lag phase is classically attributed to the duration of the requisite biochemical adaptation processes in individual cells, which are thought to switch homogeneously and responsively to the new substrate (Fig 1A and B). However, biochemical processes are inherently stochastic and cause molecule abundances to fluctuate (Elowitz *et al*, 2002). These fluctuations are often suppressed, but can also be amplified and used to generate distinct phenotypes (Balaban *et al*, 2004; Ozbudak *et al*, 2004; Choi *et al*, 2008; Losick & Desplan, 2008). Experimental (Acar

*et al*, 2008; Mitchell *et al*, 2009) and complementary theoretical studies (Kussell & Leibler, 2005) have suggested that stochastically driven systems can prevail over deterministic designs in conferring population adaptability. Although metabolism continuously adapts to unpredictable environmental changes, and certain substrate-uptake pathways have been found to exhibit phenotypic bistability (Ozbudak *et al*, 2004; Acar *et al*, 2005), central carbon metabolism as a whole is thought to operate deterministically. The possible emergence of multiple coexisting phenotypes within an isogenic cell population provides an alternative, but untested, hypothesis that the apparent lag time is caused by the exclusive growth of an initially small phenotypic subpopulation (Fig 1C).

The phenotypic subpopulation could be generated in two ways. First, the cells could 'anticipate' the environmental change by stochastically switching phenotype at any time. In this case, adaptation to a new carbon source is a passive process accomplished by cells that switched phenotype prior to the environmental change (Kussell & Leibler, 2005; Acar *et al*, 2008) (Fig 1D), similar to the mechanism employed by type II persister cells (Balaban *et al*, 2004). Second, an initially homogeneous population could actively respond to the environmental change with phenotype diversification. In this case, only a stochastically generated subpopulation of cells adapts to growth under the new conditions (Fig 1E). This responsive diversification resembles the mechanism of type I persister cells, which acquire antibiotic tolerance in response to an environmental trigger (Balaban *et al*, 2004).

Using *Escherichia coli* as a model system, we set out to determine which of these adaptation strategies (the homogeneous strategy of responsive switching, or either of the two heterogeneous strategies, responsive diversification or stochastic switching) is used when changing environmental conditions requires gluconeogenic, rather than glycolytic, growth. At the molecular level, this adaptation requires a major reorganization of central carbon flow, which *E. coli* accomplishes by repressing glycolytic genes and inducing gluconeogenic genes (Kao *et al*, 2005), particularly those participating in gluconeogenic reactions (*pckA*, *maeB*, *sfcA*), the Embden-Meyerhoff pathway (Keseler *et al*, 2009) and, in the case of acetate growth, the acetate uptake pathway (*ack*, *pta*) and glyoxylate shunt (*aceBAK*) (Wolfe, 2005).

Here, combining experiment and theory, we show that an isogenic, homogeneous *Escherichia coli* population, upon a shift

1 Institute of Molecular Systems Biology, ETH Zurich, Zurich, Switzerland
2 Molecular Systems Biology, Groningen Biomolecular Sciences and Biotechnology Institute, University of Groningen, Groningen, The Netherlands
*Corresponding author. Tel: +31 50 363 8146; E-mail: m.heinemann@rug.nl
†These authors contributed equally to this work

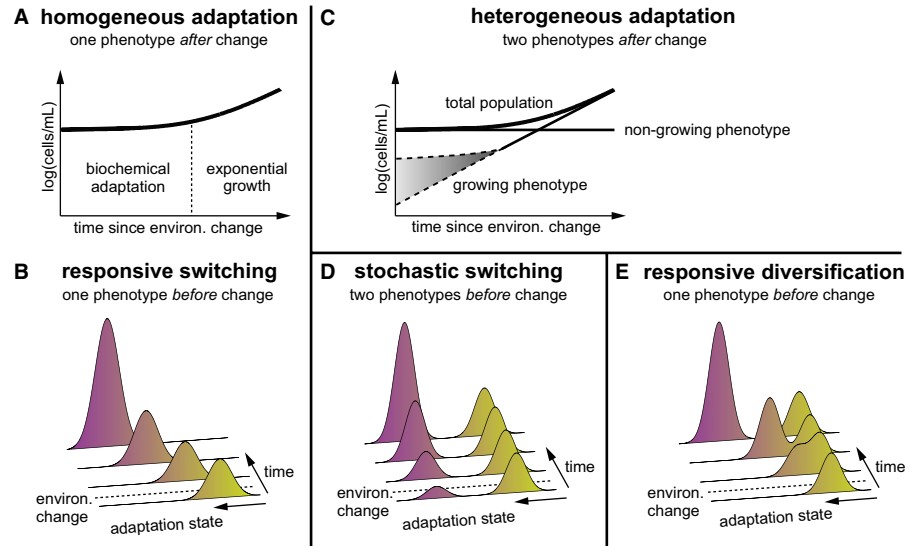

**Figure 1.   Three alternative hypotheses to explain the 'lag time' in the population growth curve after an environmental change.**

A, B    According to the classical hypothesis, a bacterial population adapts homogeneously in response to environmental change, and considerable time is required before growth resumes.

C       According to our subpopulation hypothesis, only an initially small subpopulation of cells resumes growth. Growth could immediately be at maximal rate (lower dashed line) or could increase over time (upper dashed line).

D, E    The two phenotypes can either already exist before the environmental change due to stochastically switching cells (D) or are generated from a homogeneous population in response to environmental change (responsive diversification) (E).

from glucose to various gluconeogenic carbon sources, responsively diversifies into a growing and a non-growing phenotype, causing an apparent lag time in population-level growth. Non-growing cells neglect the offered carbon source, enter a dormant state, in which they are tolerant to antibiotics, and resume growth when glycolytic conditions return. We found that the subpopulation-generating mechanism resides at the core of central metabolism, overarches the metabolic and transcriptional networks, and only allows the growth of cells achieving sufficiently high gluconeogenic metabolic flux. Thus, central metabolism does not ensure the gluconeogenic growth of individual cells, but uses a population-level adaptation resulting in responsive diversification upon nutrient changes.

## Results

### Growing and non-growing phenotypes are present after a carbon source shift

To determine whether the adaptation of *E. coli*'s central metabolism from glycolytic to gluconeogenic growth involves one or two phenotypes, we conducted substrate shift experiments from glucose to the gluconeogenic substrates acetate, fumarate, malate and succinate (Fig 2A). Just before the substrate shifts, we stained the cellular membranes with a fluorescent, membrane-intercalating dye. Because each cell's fluorescence intensity is halved with every cell division (see Supplementary Materials and Methods), this assay reports on individual cell's growth history in the new environment. We determined the distribution of fluorescence intensity in a population at multiple time points following substrate shifts (Fig 2B).

Two cellular subpopulations emerged following exposure to each tested gluconeogenic substrate; the cells of one subpopulation retained high fluorescence, because they did not divide, and cells of the other subpopulation increased in number at the same rate as their fluorescence declined, indicating growth (Fig 2B and C). We quantified the fraction of cells that managed to adapt to gluconeogenic growth ($\alpha$) as well as the steady-state growth rate of the growing cells ($\mu_g$) by fitting a model of two Gaussian distributions and of exponential growth for the growing population to the total-population fluorescence intensity distribution at multiple time points (see Supplementary Materials and Methods). For all tested substrates, at a concentration of 2 g l$^{-1}$, surprisingly, small fractions of cells adapted to gluconeogenic growth ($\alpha_{Acetate}$ = 0.5 ± 0.04, $\alpha_{Fumarate}$ = 0.001 ± 3 × 10$^{-3}$, $\alpha_{Malate}$ = 0.049 ± 0.013, $\alpha_{Succinate}$ = 0.017 ± 0.005; Supplementary Fig S1).

We excluded spontaneous suppressor mutations as a possible cause of the diversification; when cells derived from the growing subpopulation were streaked on a Luria Bertani medium plate, and single colonies were used to repeat the experiment (see Fig 2A), again two distinct growth phenotypes emerged with an identical growing population fraction $\alpha$. Also, we excluded dye toxicity as potential alternative explanation (Supplementary Fig S2A) and confirmed that the non-growing cells remain viable after carbon source shift; after transferring a glucose-adapted population to 2 g l$^{-1}$ fumarate, where only 0.1% of the cells resumed growth, we determined at multiple time points the number of viable cells by dilution plating and the total cell count using flow cytometry. At all time points, we found that the number of viable cells was equivalent to the total cell count (Supplementary Fig S3A and B). Also non-growing cells resumed growth on glucose (Supplementary Fig S3C–G).

                                                                                                

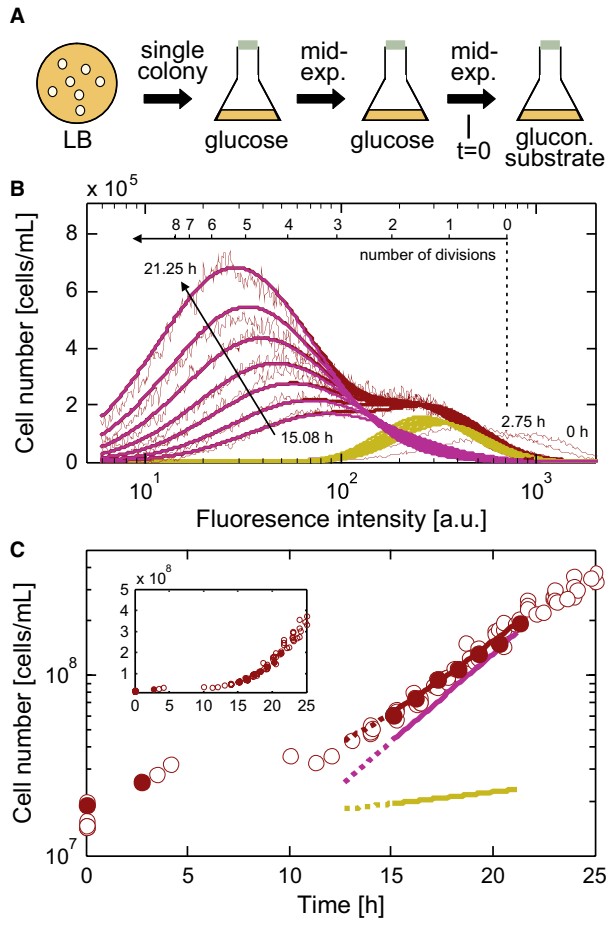

**Figure 2. Substrate shift from glucose to gluconeogenic carbon sources.**

A  Outline of the cultivation and carbon source shift procedure.

B  The distribution of fluorescence intensity at multiple time points after a shift to 0.75 g l$^{-1}$ acetate. A bi-Gaussian fit (purple, growing subpopulation; yellow, non-growing subpopulation; red, total population) reproduces the experimental data (thin lines). A fraction of non-growing cells undergoes a reductive cell division (Nystrom, 2004) causing the initial fluorescence intensity decrease; respective fluorescence distributions were not included in the fit. See Supplementary Fig S2 for validation of the staining experiment.

C  Cell count for the total population (red circles) after a shift to 0.75 g l$^{-1}$ acetate and deduced growth curves for the total population (red line), the growing subpopulation (purple line), and the non-growing subpopulation (yellow line). Yellow and purple lines represent the values of the deconvolved data shown in (B). Filled red circles indicate the time points for flow cytometric data shown in (B). The inset shows the same data with a linear *y*-axis.

To conclude, the adaptation from glucose to gluconeogenic substrates (but not from gluconeogenic substrates to glucose, see Supplementary Table S1) involves two distinct growth phenotypes of the same genotype, indicating heterogeneous adaptation. The non-growing phenotype consists of viable cells in a dormant state that resume growth when glycolytic conditions return.

## The two phenotypes are generated by responsive diversification

To discriminate between the two possible heterogeneous adaptation strategies (Fig 1D and E), we investigated whether the two phenotypes were generated at the time of the shift in carbon source (responsive diversification), or whether a phenotypic subpopulation adapted to gluconeogenic growth already existed in the glucose phase prior to the shift in carbon source (stochastic switching).

We grew *E. coli* on glucose with $^{13}$C-labeled acetate and measured $^{13}$C-enrichment patterns in the protein-bound amino acids. The mass distribution vectors of the amino acids revealed a natural labeling pattern for seven amino acids and $^{13}$C-enriched labeling pattern for eight amino acids (Supplementary Fig S4A). Mapping the amino acids to their respective precursor metabolites in central carbon metabolism (Supplementary Fig S4B) indicated that $^{13}$C-enrichment occurs only in amino acids derived from metabolites occurring below pyruvate (i.e. from the tricarboxylic acid cycle), but not in those amino acids that are derived from glycolytic intermediates above pyruvate. Therefore, next to the excretion of acetate during the glucose phase (Fuhrer *et al*, 2005), acetate is simultaneously also taken up. However, the carbon derived from acetate is only cycled through the tricarboxylic acid cycle and does not enter the Embden-Meyerhoff pathway for gluconeogenesis. A phenotypic subpopulation growing on acetate would have to perform gluconeogenesis, and thus, glycolytic intermediates above pyruvate and the amino acids derived from them would contain $^{13}$C label. As no $^{13}$C label was found in these compounds, a stochastically generated, pre-adapted subpopulation growing solely on acetate does not exist in the presence of glucose or—if obscured by the measurement uncertainty—is too small to account for the growing phenotype potentially. This result is consistent with carbon catabolite repression (Stulke & Hillen, 1999), but not with stochastic switching. Further, flow cytometry (Supplementary Fig S4C and D) uncovered no evidence of stochastic inter-phenotype switching on the gluconeogenic carbon source, indicating that the two phenotypes were stable. Thus, we concluded that two stable phenotypes were generated at the time of the carbon source shift through responsive diversification (Fig 1E).

## Subpopulation proportions depend on the carbon uptake rate

Next, we investigated whether responsive diversification is dependent on the concentration of the carbon source. We first focused on acetate, which *E. coli* produces from excess glucose. We found that the concentration of acetate in the glucose culture before the carbon source shift did not affect α (Supplementary Fig S5), and thus, the presence of acetate before the shift does not prime the cells for later acetate consumption. However, the acetate concentration after the shift determined not only the cells' growth rate μ$_g$, (with its acetate concentration-dependence following a hyperbolic Monod kinetics (Fig 3A) and kinetic parameters similar to previously published values (Wolfe, 2005)), but intriguingly also α, the fraction of adapting cells, which increased hyperbolically with the acetate concentration and leveled off at approximately 0.6 (Fig 3B). The acetate concentration-dependency of the growth rate that we observed suggests that the extracellular acetate concentration apparently influences the acetate influx and therefore also the flux into central metabolism, which let us hypothesize that also the observed α's dependency on the acetate concentration might be tied to the rate of substrate uptake.

To verify that α is indeed influenced by the substrate-uptake flux, we turned to fumarate, which—in contrast to acetate—is actively

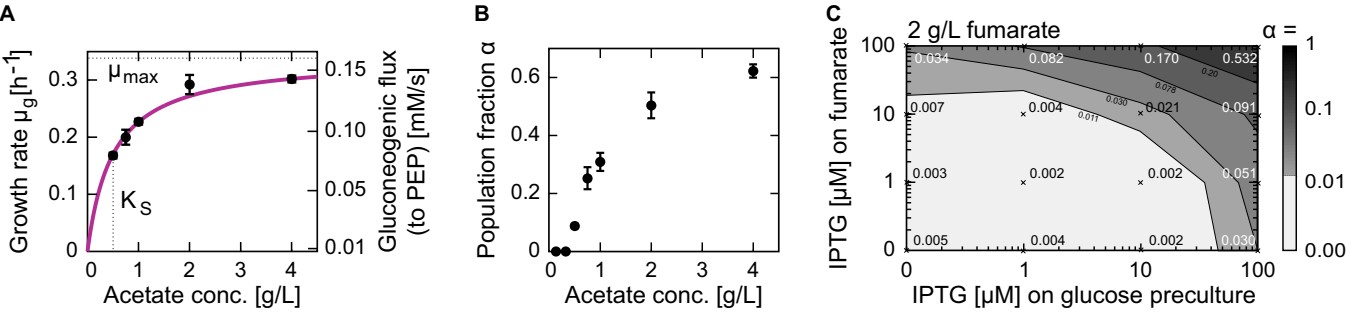

**Figure 3.  Subpopulation proportions depend on the carbon uptake rate.**

A   Dependence of the growing subpopulation's growth rate, $\mu_g$, on the acetate concentration is hyperbolic, with a maximal growth rate ($\mu_{max}$) of 0.34 h and Monod constant ($K_s$) of 0.5 g $l^{-1}$. The acetate concentrations used in these experiments are comparable to the concentrations obtained in typical glucose batch cultures (Luli & Strohl, 1990).

B   Population fraction $\alpha$ increases with acetate concentration and levels off at approximately 0.6. Error bars indicate standard deviations based on at least three replicates.

C   Population fraction $\alpha$ can be influenced by modulating the abundance of the fumarate transporter DctA when switching from glucose to 2 g $l^{-1}$ fumarate. Plasmid-based expression was induced in the wild-type strain using IPTG at different concentrations (0, 1, 10, 100 µM) on glucose (before shift), fumarate (after shift), or both. Crosses indicate conditions of individual experiments with the respective switching population fraction, $\alpha$, indicated. The steady-state fumarate uptake rates at the different induction levels 0, 1, 10, and 100 µM were 1.8, 2.1, 2.6, and 3.7 × $10^{-6}$ nmol cell$^{-1}$ h$^{-1}$ (see Supplementary Fig S6), respectively, demonstrating that the different induction levels modulate the fumarate uptake rate.

Source data are available online for this figure.

transported across the bacterial membrane by the dicarboxylate transporter DctA (Lo & Bewick, 1978). When we overexpressed this transporter in an IPTG-inducible plasmid, we confirmed that (i) the IPTG concentration could modulate the fumarate uptake rate (see Supplementary Fig S6) and found that (ii) overexpression of the transporter led to a dramatic increase in $\alpha$ (Fig 3C). While only 0.1% of uninduced cells resumed growth on fumarate, induction on both glucose and fumarate (with 100 µM IPTG) increased $\alpha$ to 53%. Induction before the substrate shift (i.e. on glucose) caused lower adapting fractions than induction after the shift (i.e. on fumarate) with the same IPTG concentrations (Fig 3C).

Thus, we concluded that $\alpha$ can be varied (i) by changing the acetate concentration in the range of its Monod constant and (ii) by modulating the expression level of the fumarate transporter. These results suggest that an increased substrate-uptake rate (for acetate accomplished by a higher acetate concentrations and for fumarate by higher transporter abundance) increases the probability that cells will resume growth on the gluconeogenic carbon source.

**Perturbations in a gluconeogenic flux sensor modulate subpopulation proportions**

We used the observation that $\alpha$ depends on the carbon uptake rate to unravel the molecular mechanism underlying responsive diversification. Recently, we argued that bacteria can infer extracellular substrate concentrations from intracellular metabolic fluxes and use these fluxes to control gene regulation (Kotte *et al*, 2010; Kochanowski *et al*, 2013), a concept that was previously demonstrated in a synthetic system (Fung *et al*, 2005). Because the mechanism responded to the rate of carbon uptake (Fig 3B and C), we focused on the flux sensors computationally predicted by Kotte *et al* (2010). One of these sensors was suggested to establish flux-dependent activity of the transcription factor Cra by binding its inhibitor, the flux-signaling metabolite fructose-1,6-bisphosphate (FBP) (Kotte *et al*, 2010). Cra is essential for growth on acetate (Chin *et al*, 1987)

and a key regulator of the glycolysis/gluconeogenesis switch (Saier *et al*, 2008); it targets almost all genes involved in the Embden-Meyerhoff pathway, the citric acid cycle and aerobic respiration (Shimada *et al*, 2011). FBP is an intermediate in the central Embden-Meyerhoff pathway and may signal the metabolic flux derived from the diverse gluconeogenic carbon sources.

First, we tested whether the Cra-FBP system indeed constitutes a flux sensor as predicted (Kotte *et al*, 2010). We used reporter plasmids to measure Cra activity for multiple acetate concentrations and, thus, multiple acetate uptake rates. We detected a strong positive correlation between gluconeogenic flux and Cra activity (Fig 4A), consistent with the prediction that the concentration of FBP, which inhibits Cra, decreases with increasing gluconeogenic flux (Kotte *et al*, 2010).

Next, we asked whether Cra activity, the output of the Cra-FBP flux sensor, is decisive for generating responsive diversification. Increasing Cra activity prior to the substrate shift (by growing cells in different glucose-limited chemostat cultures) increased $\alpha$ values (Fig 4B). We then perturbed Cra activity by individually perturbing the Cra and FBP levels. We expressed Cra using a plasmid with an IPTG-inducible promoter in a *cra* deletion mutant and found that $\alpha$ was markedly reduced in the uninduced state (49 ± 15 Cra copies) compared to the wild-type (72 ± 21 Cra copies), but the growth rate of the growing population, $\mu_g$, was not affected. When induced with 10 µM IPTG (248 ± 12 Cra copies), the wild-type $\alpha$ was restored (Fig 4C). Toward lowering the FBP concentration, we overexpressed the FBP-consuming enzyme fructose-1,6-bisphosphatase (Fbp) using a plasmid. Because we expected an increase in $\alpha$, we shifted the cells to 0.75 g $l^{-1}$ acetate and found that $\alpha$ indeed increased from the wild-type level of 0.25 ± 0.04 up to 0.70 ± 0.08 upon induction of the enzyme Fbp (Fig 4D). Toward increasing the FBP concentration in wild-type cells, we added 2-deoxyglucose-6-phosphate, a glucose-6-phosphate analogue that is not metabolized (Dietz & Heppel, 1971), but inhibits Fbp activity (see Supplementary Materials and Methods); we detected a notable decrease in $\alpha$

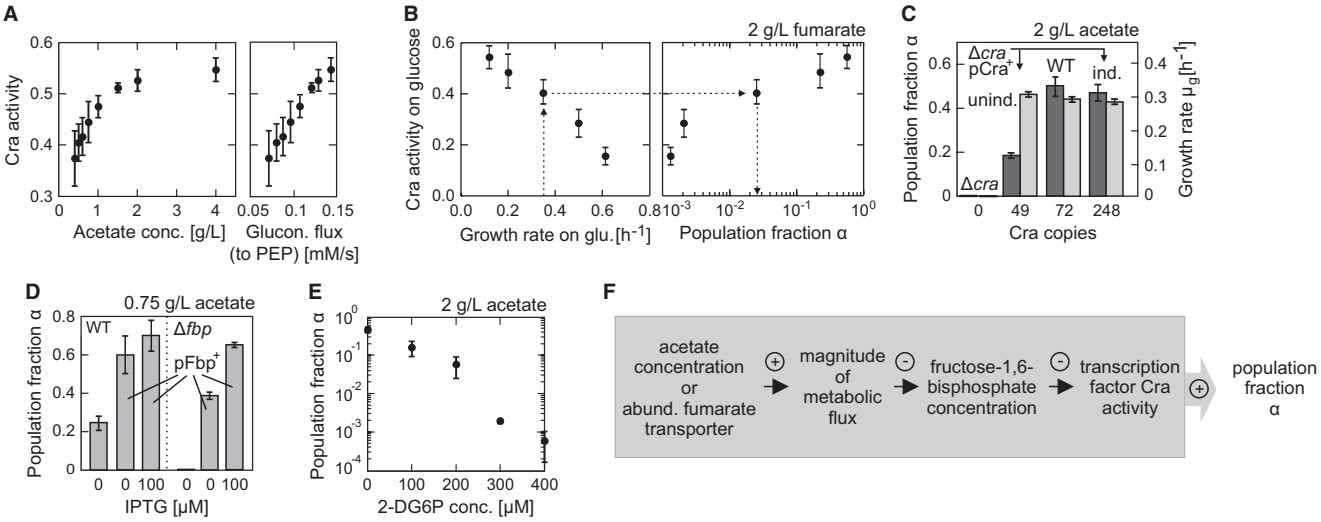

**Figure 4.  Perturbations in a gluconeogenic flux sensor modulate subpopulation proportions.**

A  The fraction of time during which Cra occupied the *pykF* promoter ('Cra activity') increased with increasing acetate concentration and thus the steady-state gluconeogenic flux to phosphoenolpyruvate (PEP). The steady-state growth rates on different acetate concentrations (see Fig 3A) and a stoichiometric metabolic network model developed by Schuetz *et al* (2007) were used to estimate the fluxes to PEP using an optimization approach ('minimization of flux' objective, see the same paper for methodology).

B  Growth rate on glucose, likely via Cra activity (left panel), influenced population fraction α (right panel) and lag phase (Supplementary Fig S7) when switching to 2 g l$^{-1}$ fumarate. The data point with the highest growth rate was from a glucose batch culture, all others are from glucose-limited chemostat cultures, in which the growth rate was controlled by the dilution rate.

C  Perturbations in the abundance of transcription factor Cra, through knockout or overexpression, affected α (dark gray bars), but not μ$_g$ (light gray bars).

D  The population fraction increased above wild-type levels, in both wild-type (left set of bars) and Δ*fbp* (right set of bars), when fructose-1,6-bisphosphatase (Fbp) was overexpressed from an inducible plasmid.

E  The population fraction α decreased with an increasing concentration of 2-deoxyglucose-6-phosphate (2-DG6P), a non-metabolized glucose-6-phosphate analogue that inhibits the enzyme Fbp (see Supplementary Materials and Methods).

F  Summary of the critical interactions that determine population fraction α. A plus or minus indicates a positive or negative relationship of the interaction.

Data information: Error bars indicate standard deviations based on at least three replicates.

Source data are available online for this figure.

(Fig 4E). Together with the observation that other potentially involved transcription factors (such as Crp, ArcA and IclR) were found to have no role in the generation of the bistability (see Supplementary Table S2), we conclude that carbon uptake flux is measured by the Cra-FBP flux sensor and is decisive for generating responsive diversification (Fig 4F).

**Bistable phenotypes are generated by central metabolic regulation**

Next, we aimed to uncover the molecular system that generates the two distinct phenotypes. This system certainly involves the Cra-FBP flux sensor. Notably, flux-dependent Cra activity regulates the production of many enzymes that catalyze gluconeogenic reactions upstream of FBP formation, and therefore, the carried gluconeogenic flux to FBP could close a global feedback loop overarching the metabolic and transcriptional networks (Fig 5A). Because positive feedback architectures can propagate stochastic fluctuations into multiple phenotypes (Smits *et al*, 2006), we investigated whether the Cra loop is indeed capable of generating responsive diversification.

To help understand this rather complex system, we resorted to the descriptive power of a mathematical model. We thus constructed a differential equation model (see Supplementary

Materials and Methods) that combines the reaction sequence from gluconeogenic substrate uptake to FBP formation into a single reaction catalyzed by a fictitious super-enzyme E. The production of E is activated by Cra (for the effect of more nuanced regulation, see Supplementary Materials and Methods), and the activity of Cra is inhibited by FBP. One way to make such a feedback loop positive is if FBP levels drop with increasing flux thereby introducing a second negative sign into the feedback loop (note, the first negative sign is introduced by the inhibition of Cra by FBP), such that the sign of the overall feedback loop becomes positive and the generation of two coexisting phenotypes possible. Through a metabolomics experiment, we verified that FBP levels indeed decrease with increasing gluconeogenic flux (Fig 5B).

Next, we investigated how the flux-FBP level relationship could be inverted. While we found that the Fbp expression levels are constant across the employed acetate concentrations (between 0.5 and 2 g l$^{-1}$ acetate; *P*-value 0.86), Fbp's strong allosteric activator PEP (Hines *et al*, 2006; Link *et al*, 2013) showed higher concentrations at the higher flux conditions (Fig 5C) suggesting that this flux-dependent feed-forward activation of Fbp activity on the allosteric level could invert the flux-FBP level relationship. If strong enough, the flux-dependent allosteric feed-forward activation of Fbp's activity by PEP turns the sign of the global Cra feedback loop positive

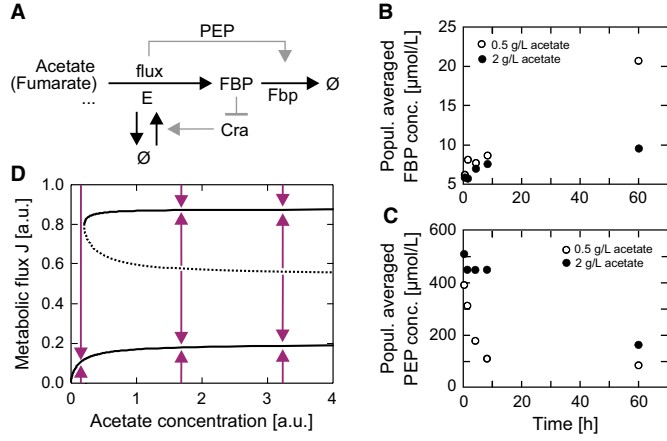

**Figure 5. Model of the bistability-generating circuit.**

A    Model of the bistability-generating circuit. E denotes a fictitious super-enzyme catalyzing combined reactions. The enzyme fructose-1,6-bisphosphatase (Fbp) is activated in a flux-dependent manner. The metabolite fructose-1,6-bisphosphate (FBP) represses E production by inhibiting E's transcriptional activator Cra. PEP, phosphoenolpyruvate.

B, C  Population-averaged metabolite levels after a switch from glucose to acetate (0.5 g $l^{-1}$, open symbols; 2 g $l^{-1}$ closed symbols) showed that at the higher acetate concentration—and thus at the higher gluconeogenic flux condition—FBP levels are lower (B), while the level of the strong allosteric activator of the Fbp enzyme, phosphoenolpyruvate (PEP), is significantly higher at increased fluxes, accomplishing a flux-dependent feed-forward activation of the Fbp enzyme (C).

D    Bifurcation diagram of metabolic steady-state fluxes J as a function of the extracellular acetate concentration. The system is capable of expressing two stable steady-state fluxes (bold lines) and one unstable steady-state flux (dashed line), which acts as a watershed separating the convergence regions of the high (growing) and low (non-growing) stable steady states. Arrows show the direction of system dynamics.

Source data are available online for this figure.

and allows for the generation of two distinct phenotypes from a unimodal distribution. In the model, we included a cooperative flux-dependent activation of the enzyme Fbp (see Supplementary Materials and Methods).

The separation of a homogeneous population into two subpopulations is illustrated by a bifurcation analysis of the mathematical model performed in the bistable regime (see Supplementary Materials and Methods). Two stable metabolic fluxes emerge, which can be interpreted as the growing and non-growing phenotypes, and which are separated by a watershed in the bifurcation diagram (Fig 5D, Supplementary Fig S8). If, after a substrate shift, a particular cell immediately achieves a gluconeogenic flux above the watershed, the system dynamics drag it toward the high steady state and it adopts the growing phenotype; otherwise, it approaches the low steady state and adopts the non-growing phenotype. This system requires a capacity for immediate utilization of gluconeogenic substrates upon glucose removal, which was demonstrated by the $^{13}$C tracer experiments (Supplementary Fig S4A and B). A unimodal population separates into these two coexisting phenotypes when cell-to-cell variation in the abundance of the super-enzyme E, which is encouraged by the very low copy number of Cra (see Fig 4C), causes E's established flux to exceed the watershed in some cells but not in others. After separation, additional state-stabilizing regulatory adjustments likely occur in non-growing, and possibly

growing, cells, but these are not covered by the model. Differences between the cells' capability to generate flux to FBP after the carbon source shift effectively cause the bifurcation. Flux bottlenecks are likely generated by stochastic differences in gluconeogenic enzyme expression, with the bottlenecks eventually residing in each individual cell at different reactions.

The model's system properties were consistent with our experimental observations. Increasing acetate concentrations in the range of the Monod constant and a greater abundance of the fumarate transporter resulted in higher gluconeogenic uptake fluxes, shifting cells above the watershed and increasing α (Fig 3A–C). The carbon flux derived from all tested gluconeogenic substrates is funneled into the Embden-Meyerhoff pathway; therefore, the mechanism described in the model can explain responsive diversification for all tested substrates. Further, mathematical model analysis (see Supplementary Materials and Methods) predicted that a decrease in the production rate of E would increase the watershed level, which would substantially reduce α but, counter-intuitively, only marginally affect μ$_g$, as the flux of the high steady state would basically remain unaltered (Fig 6A). One attempt to lower the 'production rate' of the fictitious super-enzyme E in a concrete experiment is to knock out isoenzymes (*acnA, acnB*) or parallel pathways (*maeBsfcA, ppsA, pckA*; Fig 6B) subsumed into E. Besides potentially introducing metabolic flux bottlenecks, these knockouts might also introduce bottlenecks in the 'production rate' of the fictitious super-enzyme E. Following the substrate shift, several of these mutants (*ppsA, maeBsfcA, acnB*) exhibited a marginally reduced μ$_g$ and drastically reduced α (Fig 6C), suggesting the introduction of a transcriptional but not metabolic bottleneck; wild-type-like values were restored when the alternative isoenzyme or parallel pathway was overexpressed (Supplementary Fig S9). Suppressor mutants were excluded in strains that showed the long lag phase (Supplementary Table S1).

Taken together, our experimental findings and the model-based integration of these findings in a system context strongly suggest that the molecular system shown in Fig 5A is the mechanism responsible for the generation of responsive diversification upon carbon source shifts.

## Cells of the non-growing phenotype are dormant persisters

Lastly, we asked whether the non-growing dormant cells that occur after the carbon source shift resemble persister cells tolerant to antibiotics. Persister cells are dormant variants of regular cells that can survive treatment with antibiotics, but remain sensitive to that antibiotic upon being regrown (Lewis, 2010). To test for our non-growing cells' antibiotic tolerance, we subjected cells from several time points after a glucose-to-fumarate shift to different antibiotics for 2 h and determined cell viability by plating. Using antibiotics that kill growing cells, specifically ampicillin, kanamycin, and rifampicin, we found that 86 to 97% of the non-growing cells tolerated the treatment (Supplementary Fig S10A–D). Using antibiotics that also kill non-growing cells, specifically ofloxacin and mitomycin C, we detected that 18% of the non-growing cells tolerated treatment (Supplementary Fig S10E and F). Cells exponentially growing on glucose did not survive identical antibiotic treatment. We excluded that the tolerance exhibited by the non-growing cells was due to genetically acquired resistance; when cells that survived antibiotic

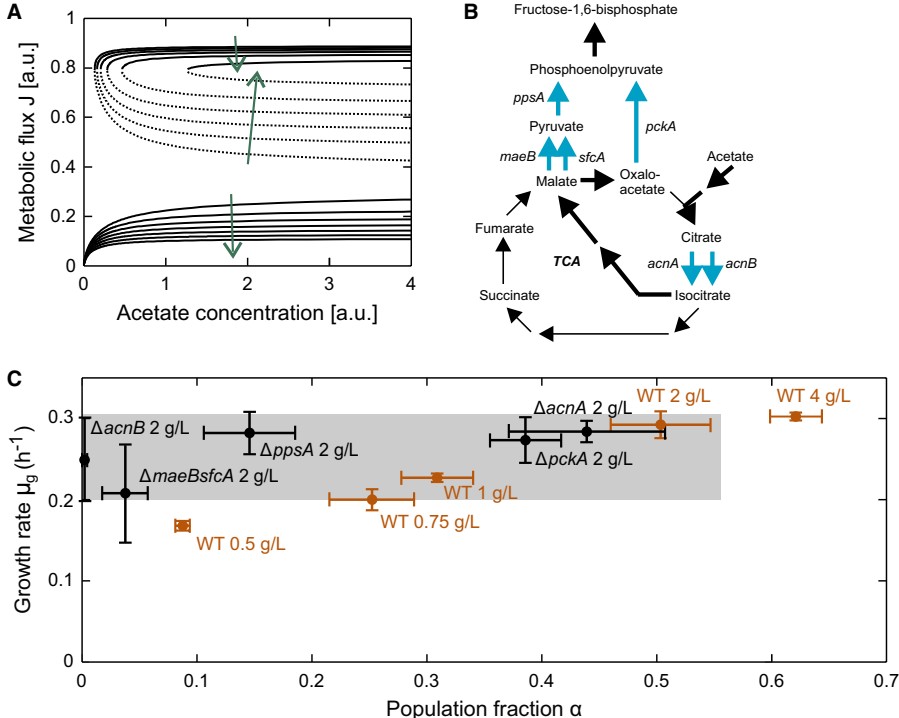

**Figure 6. Perturbations to the model and experimental validation.**

A  Bifurcation diagrams for different super-enzyme E production rates. Arrows indicate the direction of decreasing E production rates with which the convergence region of the growing phenotype gradually decreases and that of the non-growing phenotype increases (and α decreases), and steady-state flux J and the growth rate of the growing phenotype basically remain stable.

B  Reactions catalyzed by super-enzyme E. Bold arrows indicate the route of major carbon flux from acetate to FBP. Thin arrows complete the citric acid cycle. Blue arrows highlight isoenzymes and parallel pathways that are knockout targets for experimentally introduced different E production rates.

C  Growing population fraction α and its growth rate $\mu_g$ for different strains when switching to acetate with the indicated concentrations. Orange data points indicate wild-type behavior in which both α and $\mu_g$ increase with increasing acetate concentration. Black points indicate the behavior of knockout mutants with potential different E production rates. In the Δ*acnB*, Δ*ppsA*, and Δ*maeBsfcA* deletion strains, α is markedly reduced, whereas the Δ*acnA* and Δ*pckA* mutants exhibit nearly wild-type behavior. Error bars indicate standard deviations based on at least three replicates.

Source data are available online for this figure.

exposure were regrown on glucose minimal medium, renewed antibiotic exposure killed all of the cells. Thus, the cells of the non-growing phenotype, largely tolerant against antibiotics, resemble type I persister cells (Balaban *et al*, 2004), and here, we show that persister cells can arise from insufficient carbon uptake rates, for example, occurring after shifts in nutrient availability.

# Discussion

Our data show that, upon a shift from glucose to various gluconeogenic carbon sources, an isogenic, homogeneous cell population responsively diversifies into growing and non-growing phenotypes, causing an apparent lag time in population-level growth. The responsible molecular mechanism resides at the core of central metabolism. Metabolic flux is used as a controlling factor, and a cell grows on gluconeogenic substrates only if it achieves a gluconeogenic flux above a crucial watershed. Non-growing cells neglect the offered carbon source and enter a protective dormant state in which they are, for example, tolerant to antibiotics and resume growing when glycolytic conditions return.

## Responsive diversification offers explanation for the phenomenon of lag phases

Lag phases are well known characteristics of almost every bacterial cell culture and were traditionally attributed to the duration of necessary biochemical adaptation processes. However, this theory fails to explain long lag phases, and thus, these lag phases are often ascribed to spontaneous mutations (Egler *et al*, 2005; Takeno *et al*, 2010).

For the first time, we offer a consistent explanation for the phenomenon of lag phases. Although the time needed for biochemical adaptation may still contribute to lag phases, we showed that lag time is largely caused by the exclusive growth of an initially small subpopulation (see Supplementary Fig S7), and we found that this also generalizes to other carbon source shifts (see Supplementary Table S1). Colleagues from the Netherlands recently confirmed similar behavior in yeast (van Heerden *et al*, 2014) and *L. lactis* (Solopova *et al*, 2014). With knowledge of the respective diversification-generating mechanism, wild-type lag phases can be shortened, as we demonstrated for fumarate (Fig 3C) and acetate (Fig 4C and D) with broad implications for microbiology and biotechnology.

    

### Flux-induced phenotypic bistability generalizes to central metabolism

Not being specific to a particular gluconeogenic substrate, the molecular mechanism that generates phenotypic bistability in response to a glucose-to-gluconeogenic substrate shift is general in nature and resides at the core of central metabolism. Thus, phenotypic bistability, which has so far only been observed in a few specific substrate-uptake pathways (Ozbudak *et al*, 2004; Acar *et al*, 2005), generalizes to central metabolism as a whole. Such bistable phenotypes ultimately arise from stochastic variability in biomolecule abundance and are generated through positive feedback (Balazsi *et al*, 2011). A crucial element of the here uncovered feedback loop is the metabolic flux that connects carbon uptake to the level of an intracellular, flux-signaling metabolite (i.e. FBP). As the previously observed occurrences of bistability within substrate-uptake pathways (Ozbudak *et al*, 2004; Acar *et al*, 2005) also involve flux sensor motifs (Kotte *et al*, 2010), flux-induced bistability may be a ubiquitous principle of metabolism.

### Responsive diversification manifests a trade-off of cellular regulation

Our study reveals that, after glucose depletion, surprisingly few cells accomplish the transition to gluconeogenic growth. In fact, as Fbp overexpression increases α in the wild type (Fig 4D), *E. coli*'s adaptation to gluconeogenic substrates is not optimal at the single-cell level. Why did such behavior evolve? Our results indicate that fast glycolytic growth correlates negatively with α (Fig 4B). Only cells that refrain from growing very fast on glucose maintain their ability to switch to gluconeogenic growth upon glucose depletion; cells with a very fast glycolytic growth rate cannot generate sufficiently high initial gluconeogenic flux through the Embden-Meyerhoff pathway and must enter dormancy. This observation indicates an inherent trade-off between the rate of glycolytic growth and the ability to adapt to gluconeogenic growth, which is in agreement with a recently suggested trade-off between optimality under one given condition and minimal adjustment between conditions (Schuetz *et al*, 2012).

Thus, our results demonstrate that *E. coli*'s adaptation to fluctuating carbon sources is not optimized at the single-cell level, but at the population level eventually through conditional bet-hedging (Veening *et al*, 2008; Beaumont *et al*, 2009; de Jong *et al*, 2011). Cells could already hedge their bets on glucose and balance their glycolytic growth rate with their ability to adapt to gluconeogenic growth. The population is unimodal on glucose, and each cell is placed stochastically and gradually in a landscape between these two competing objectives. Thus, the population could cope with the present trade-off by sampling a large space of possible expression patterns. The conditional bet-hedging becomes apparent only after glucose depletion, when a certain level of adaptability to gluconeogenic growth is suddenly needed; thus, responsive diversification is observed.

### Limited carbon influx is a prime trigger of persistence

Finally, our data indicate that a threshold carbon uptake rate exists, below which rate cells enter dormancy and become largely tolerant

to a number of antibiotics. Upon the cessation of glycolytic growth conditions, a flux-induced mechanism controls whether cells become dormant. This finding could explain why the frequency of persister cells increases during entry into the stationary phase (Gefen *et al*, 2008; Luidalepp *et al*, 2011). An intriguing question is whether the small population that occurs naturally and makes up the basis of persistence (Balaban *et al*, 2004; Schumacher *et al*, 2009) is also triggered by (stochastically) limited carbon uptake rates.

## Materials and Methods

### Strains and plasmids

*Escherichia coli* K12 strain BW25113 served as the wild-type strain. Gene deletions were transferred to the wild-type background from deletion strains of the Keio collection (Baba *et al*, 2006) using P1 phage transduction (Mitchell *et al*, 2009) and verified by colony PCR. The IPTG-inducible over-expression plasmids pP$_{tac}$-cra, pP$_{tac}$-acnA, pP$_{tac}$-fbp as well as the reporter plasmid pP$_{pykF}$-gfp were obtained from available libraries (Kitagawa *et al*, 2005; Saka *et al*, 2005; Zaslaver *et al*, 2006). The reporter plasmid pP$_{fbp}$-gfp was constructed in analogy to the plasmids in the library by (Zaslaver *et al*, 2006). A de-regulated *pykF* reporter plasmid variant, pP$_{pykF*}$-gfp, was constructed by mutating the Cra binding site via PCR. See Supplementary Materials and Methods for details on strains and plasmids.

### Growth media

M9 medium with 1.5 g l$^{-1}$ (NH$_4$)$_2$SO$_4$, and 1 mg l$^{-1}$ thiamin-HCl as the sole vitamin, was used. Carbon sources were added from stock solutions adjusted to pH 7.

### Cultivation

Pre-cultures from single colonies were grown overnight in M9 plus 5 g l$^{-1}$ glucose. Respectively required antibiotics were added to pre-cultures with strains carrying plasmids (ampicillin, 20 µg ml$^{-1}$; kanamycin, 25 µg ml$^{-1}$). Cells were re-inoculated into M9 medium plus 5 g l$^{-1}$ glucose. For batch cultures, cells were cultivated in 500-ml Erlenmeyer flasks containing 50 ml growth medium (300 rpm, 37°C). For chemostat cultures, cells were grown in mini-chemostats as described by Nanchen *et al* (2006) on M9 medium plus 1 g l$^{-1}$ glucose.

### Carbon source switching experiments and cell staining

A total of $1.5 \times 10^9$ cells from an exponentially growing culture, or a chemostat culture, was harvested in mid-exponential phase (OD$_{600} \approx 0.5$) and centrifuged for 4 min at 4,000 *g* and 4°C. The supernatant was discarded, the cells washed or stained, and inoculated (~$2 \times 10^7$ cells ml$^{-1}$) into M9 medium plus acetate, succinate, fumarate, or malate. For washing, the cells were resuspended in 5 ml ice-cold M9 medium, centrifuged a second time, and re-suspended in 1 ml of room temperature M9 medium. For staining, the cells were resuspended in 500 µl of room temperature dilution

   

buffer C (Sigma-Aldrich). A freshly prepared mixture of 10 μl PKH26 or PKH67 dye (Sigma-Aldrich) and 500 μl dilution of buffer C (both at room temperature) was added. After 3 min at room temperature, 4 ml ice-cold filtered M9 medium containing 1% (w/v) bovine serum albumin (AppliChem) was added. After centrifugation, the supernatant was discarded and the cells washed twice.

### Flow cytometry

Samples were diluted with M9 medium to an $OD_{600}$ of 0.001. Briefly, vortexed samples were analyzed with either a BD Accuri C6 flow cytometer (BD Biosciences; 20 μl, flow rate: medium, FSC-H threshold: 8000, SSC-H threshold: 500) or a FACS Calibur flow cytometer (BD Biosciences; 1 min, no gating, flow rate: high, FSC: E02, SSC: 327, FL-1: 999, FL-2: 700: all log, primary: SSC, threshold: 50), with gating performed in FlowJo 8.2 (Tree Star). To determine the absolute cell counts with the latter instrument, 20 μl of vortexed counting beads (CountBright, Invitrogen) was added to a 380 μl cell suspension prior to analysis.

### $^{13}$C labeling experiment

To identify whether an acetate-growing phenotypic subpopulation already exists in the glucose growth phase, cells were grown in M9-glucose pre-culture, transferred to two M9-glucose (5 g $l^{-1}$) main cultures supplemented with either 1 g $l^{-1}$ unlabeled (i.e. naturally labeled) or 1 g $l^{-1}$ fully $^{13}$C-labeled acetate. Samples were taken in the mid-exponential growth phase ($OD_{600}$ = 1.2) from culture with the labeled and unlabeled acetate, and a positive control sample was taken in the stationary phase ($OD_{600}$ = 3.3) from the culture with the labeled acetate. At this point, the labeled acetate was taken up. Samples were processed according to (Ruhl *et al*, 2010) and labeling patterns in the protein-bound acids analyzed by GC-MS (Zamboni *et al*, 2009).

### Fructose-bisphosphatase (Fbp) and Cra overexpression

The wild-type strain, the *fbp* deletion mutant, the *fbp* deletion mutant harboring the IPTG-inducible plasmid pP$_{tac}$-fbp, and the *cra* deletion mutant carrying the IPTG-inducible plasmid pP$_{tac}$-cra were grown in M9 medium plus glucose (5 g $l^{-1}$) with 0 or 10 μM IPTG, then transferred to M9 medium plus acetate (0.75 g $l^{-1}$ for Δ*fbp*, 2 g $l^{-1}$ for Δ*cra*) with the same amount of IPTG.

### Inhibition of Fbp activity

Wild-type cells were grown in M9 medium plus glucose (5 g $l^{-1}$), stained, and transferred to M9 medium plus acetate (2 g $l^{-1}$) with varying concentrations of 2-deoxy-glucose-6-phosphate (2DG6P).

### Estimation of Fbp abundance

Wild-type cells harboring the pP$_{fbp}$-gfp plasmid were adapted to growth on M9 medium plus acetate (0.5 and 2 g $l^{-1}$) and grown at steady state. Steady-state growth was achieved by using low cell concentrations (< 5 × 10$^9$ cells $l^{-1}$) that do not cause a significant change in the carbon source concentration within the measurement window. Cellular fluorescence determined by microscopy

was background corrected and used as a proxy for protein abundance.

### Perturbation of fumarate transporter abundance

The wild-type strain harboring the IPTG-inducible plasmid pP$_{tac}$-dctA was grown in M9 medium plus glucose (5 g $l^{-1}$) with 0, 1, 10, or 100 μM IPTG, then stained, and transferred to M9 medium plus fumarate (2 g $l^{-1}$) with 0, 1, 10, or 100 μM IPTG. Uptake rates at different IPTG levels were determined in fumarate-adapted cultures, where cell counts were determined at different time points by flow cytometry, and extracellular fumarate concentrations with HPLC. Dynamic cell counts and fumarate concentration data were fitted (MCMC toolbox for Matlab) to a model describing fumarate depletion and cell growth with the growth rate following Monod kinetics.

### Quantification of Cra abundance

The wild-type and *cra* deletion mutant harboring the IPTG-inducible plasmid pP$_{tac}$-cra were grown in M9 medium plus glucose (5 g $l^{-1}$) or M9 plus acetate (0.75 g $l^{-1}$) plus 0 or 10 μM IPTG. Harvested cells were washed and lysed, and the extracted proteins were digested with trypsin as described previously (Malmstrom *et al*, 2009). Then, 10 pmol of heavy labeled reference peptide was added to the digests, each containing 100 μg of total peptide. After desalting the peptides with macro-spin columns (Harvard Apparatus), an aliquot containing 1 μg of peptide was subjected to targeted mass spectrometry using previously specified instrument settings (Picotti *et al*, 2009). The total number of cells in the samples was determined by flow cytometry. Averages and standard deviations were derived from glucose experiments for the wild-type cells and from glucose and acetate experiments for other strains.

### Quantification of Cra activity

Wild-type cells, wild-type cells harboring the pP$_{pykF}$-gfp plasmid, and wild-type cells carrying the pP$_{pykF}$*-gfp plasmid (a variant with the Cra binding site removed) were adapted to a range of acetate concentrations, transferred to a plate reader (TECAN Infinite Pro 200), and grown at the steady state. Cra activity was calculated as 1 (promoter activity *pykF*$_{regulated}$/promoter activity *pykF*$_{deregulated}$) (Bintu *et al*, 2005). For batch cultures, promoter activities for the regulated and the deregulated *pykF* promoter were calculated as dGFP/dt/OD during exponential growth. For chemostat cultures, promoter activities were calculated as D × GFP/OD, where D denotes the dilution rate. Correction for background fluorescence for batch and chemostat cultures was performed using a strain bearing a promoterless GFP reporter plasmid.

### Quantification of FBP and PEP levels

Cells were grown on glucose to an $OD_{600}$ = 0.5, washed, and shifted to either 0.5 or 2 g $l^{-1}$ acetate at an $OD_{600}$ = 0.25. The experiment was carried out in a chamber heated to 37°C, and the cultures were mixed with a magnetic stirrer. Samples (3 ml) were taken and processed by fast filtration of as described (Bolten *et al*, 2007) without additional washing of cells. Extraction was carried out by immediately placing the filter in 4 ml of aqueous 50% (v/v) ethanol

solution heated to 78°C and incubation for 3 min at 78°C. The extraction solution transferred to two 2-ml tubes was centrifuged for 2 min at −15°C at 15,000 $g$ to remove cell debris. The samples were dried (Christ RVC 2-33 CD centrifuge and Christ Alpha 2–4 CD freeze dryer) and resuspended in water. Metabolites were identified and quantified by ultrahigh performance liquid chromatography combined with mass spectroscopy (Thermo TSQ Quantum Ultra triple quadrupole instrument, Thermo Fisher Scientific) as described before (Buescher *et al*, 2010). Conversion of the OD-normalized metabolite measurements to cellular concentrations was done using the relationship between total cell volume and OD as described before (Volkmer & Heinemann, 2011).

**Supplementary information** for this article is available online: http://msb.embopress.org

## Acknowledgements

We thank M. Rottmar, J. Zaugg, M. Berney, and M. Ackermann for early work and discussions; A. Schmidt, M. Rühl, K. Mitosch, S. Vedelaar, and G. Zampar for supporting experiments; K. Kochanowski for supporting experiments and for generating the reporter plasmids; S. Panke and U. Sauer for support and discussions; and B. Poolman, A. van Ojien, J.W. Veening, O. Kuipers, and U. Sauer and members of the Heinemann laboratory for critically reviewing the manuscript. This work was supported by an ETH Zurich grant (INIT-SCM), a Roche Research Foundation grant, an NWO-Vidi grant, a DuPont Young Professorship Award (all to MH), and the Groningen Centre for Synthetic Biology.

## Author contributions

OK, BV, and MH conceived and designed the study. BV and JR conducted the experiments and analyzed data. OK developed, implemented, and analyzed the mathematical models. OK and MH wrote the manuscript with input from all authors.

## Conflict of interest

The authors declare that they have no conflict of interest.

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
