## [Review Process File · Molecular Systems Biology]

Phenotypic bistability in *Escherichia coli*'s central carbon metabolism

Oliver Kotte, Benjamin Volkmer, Jakub L. Radzikowski and Matthias Heinemann

Corresponding author: Matthias Heinemann, University of Groningen

Review timeline:

Submission date:	27 November 2013
Editorial Decision:	15 January 2014
Revision received:	17 April 2014
Editorial Decision:	14 May 2014
Revision received:	22 May 2014
Accepted:	23 May 2014

Editor: Thomas Lemberger

Transaction Report:

1st Editorial Decision

15 January 2014

Thank you again for submitting your work to *Molecular Systems Biology*. We have now heard back from the two referees who agreed to evaluate your manuscript. As you will see from the reports below, the referees find the topic of your study of potential interest. They raise, however, concerns on your work, which should be addressed in a major revision of the manuscript.

The reviewers find the main observations interesting and novel. Without repeating all the points listed in their reports, the reviewers raise concerns with regard to three aspects of the study:

- The nature of the stochastic process that underlies the generation of the two phenotypic sub-populations. While identifying the exact source of stochasticity might be difficult (even though we note that reviewer #2 provides constructive suggestions), a better discussion of this point is required.
- The relationship between the fraction of growing cells (α) and the length of the resulting (apparent) lag phase should be clarified (see points raised by reviewer #2).
- The most substantial concerns refer to the proposed mechanisms that would explain the observed responsive phenotypic bistability--in particular the increase in gluconeogenic flux but drop in FBP levels that creates the double negative feedback loop via Cra. While measuring ALL enzymatic activities might be challenging, more direct experimental evidence for the changes in glucogenogenic fluxes and for the impact of PEP on Fbp activity seems to be important, as noted by reviewer #1.

On a more editorial level, we would strongly encourage you to provide the source data for key quantitative measurements shown in the main figures. These data can be submitted as 'source data files' associated to figure panels, so that interested readers can download them directly from the figures for the purpose of re-analysis, visualization or integration in other works (see <http://msb.embopress.org/authorguide#a3.4.3>).

If you feel you can satisfactorily deal with these points and those listed by the referees, you may wish to submit a revised version of your manuscript. Please attach a covering letter giving details of the way in which you have handled each of the points raised by the referees.

REFeree REPORTS:

Reviewer #1:

SUMMARY

The authors have written a paper describing a very intriguing and comprehensive study which demonstrates phenotypic bistability in *Escherichia coli* central carbon metabolism in response to a shift from glycolytic to gluconeogenic growth, and elucidates the nature of this phenomenon. The study explored how isogenic populations of bacterial cells are able to produce alternative phenotypes through stochastic means, conferring upon the population a degree of adaptability to environmental fluctuations. The study presents a diverse array of experiments which characterise the stochastic growth responses of these cells including: fluorescence staining, flow cytometry, LC/MS, and a host of studies with mutants and cells containing inducibly expressed genes and reporter plasmids.

The primary conclusions of this study were that the lag in growth observed in *E. coli* in response to a shift from glycolytic to gluconeogenic substrates can be attributed to a responsive stochastic diversification of an isogenic population of cells with a shared phenotype into two sub-populations with distinct and stable growth rates. Additionally, it was shown that the proportion of cells switching to the high growth rate phenotype was dependent upon gluconeogenic substrate concentration, and by implication, on gluconeogenic flux and the activity of the global transcription factor Cra. A mechanism was proposed for the bistable diversification of this system involving Cra and fructose 1,6-bisphosphate (FBP), and explored by the construction and scrutinizing of a simplified core model. On this basis, the authors speculate that it is indeed possible for the observed bistability to be achieved by a feed forward effect between a flux-dependent gluconeogenic metabolite concentration and its allosteric activation of the FBP consuming fructose 1,6-bisphosphatase (FBPase) reaction, which results in the the alleviation of the inhibition that FBP exerts on Cra.

GENERAL REMARKS

The authors present a well-rounded experimental study involving a varied array of evidence. The conclusions drawn from the experimental evidence appear to be valid. This paper further elucidates the nuanced mechanisms involved in the regulation of glycolytic and gluconeogenic metabolism, and this discovery has implications for, amongst others, industry, where the presence of growing and non-growing phenotypes are an important concern as *E. coli* is a favoured protein expression vector; and for medicine, as the discussion about antibiotic sensitivity demonstrated.

The intention of the authors to provide a putative mechanistic explanation for the observed experimental behaviour by constructing a mathematical model of the system is praiseworthy. However, the constructed model was oversimplified and in our opinion the conclusions drawn with regard to the proposed bistability mechanism are not fully justified.

MAJOR CRITICISMS

The core model developed in the latter sections of the paper is too simplistic and we found the conclusions drawn from this model to be too speculative. Some weaknesses of the model are:

1. It is not necessarily true that the gluconeogenic substrate X maintains a simple proportional relationship to flux. Correlation does not imply causation, and moreover the gluconeogenic flux was not measured directly but calculated on the basis of an FBA model (Fig. 4A). The underlying mechanistic causes may well lie somewhere else. For example, a number of regulatory relationships span the central metabolic pathways in *E. coli*, often incorporating the energy charge and redox state

of the cell; facets that were excluded from this model.

2. It is true that with reduced activities of the enzymes catalysing the reactions from PEP to FBP, in a simple pathway a higher concentration of PEP would be required to achieve the same flux (p. 35, Supplementary Information). However, the extent of this required increase in PEP depends on how far the reactions are from equilibrium, and if there is sufficient excess capacity, the same steady-state flux could in principle be maintained by a virtually unaltered PEP concentration even if enzyme levels are decreased substantially. In general, the control of the steady state concentration of any metabolite in a network is shared amongst the members of the network, and close-to-equilibrium reactions tend to have low control coefficients.

3. Since a detailed kinetic model of *E. coli* central carbon metabolism has been published recently (Peskov et al. 2012, FEBS Journal), it should be possible to produce a quantitative model of all the molecular interactions involved in central carbon metabolism to identify the actual underlying molecular mechanisms. Since PEP exists at a complex regulatory node in central carbon metabolism, its treatment in the current model is too superficial.

4. The underlying molecular mechanism can be further elucidated by measuring the activities of all the enzymes in the gluconeogenic pathway from acetate to FBP using classical biochemical activity assays (methods have been described) to provide a better quantitative picture of the metabolic reprogramming that occurs. This should be done at $t=0$ directly after the switch, and at $t=15$ h when the majority of the culture would be made up of fast-growing cells. This would prove conclusively which gluconeogenic enzymes are being upregulated and to what extent.

5. It is not clear in the literature whether PEP acts as an activator (Hines 2006) or inhibitor (Babul 1983) of the FBPase reaction (or whether a more complex allosteric relationship exists). This should be clarified experimentally as it is fundamental to the authors' hypothesis.

6. In conclusion, the experimental evidence for responsive diversification into two separate subpopulations upon switch to a gluconeogenic substrate is convincing, as is the requirement for Cra and Fbp to observe this effect. However, the core model put forward is too speculative to provide convincing evidence for the hypothesised mechanism. To be convincing, the hypothesis should be tested using a detailed quantitative molecular-level model of central carbon metabolism in *E. coli*. In addition, techniques such as metabolic control analysis can provide insight into the control and regulation of fluxes and metabolite concentrations under the different conditions.

MINOR CRITICISMS

1. A number of grammatical errors were discovered in the text, and the quality of the language use in general could be improved. A (non-exhaustive) list of examples:
 - the verb "describing" in the middle of page 5 seems inappropriate
 - "rationale", not "rational"
 - "kinetics", not "kinetic"

2. The figure reference to Fig. 4B, D (p. 13, 6 lines up) is wrong. Figure cross-references should be re-checked.

Reviewer #2:

Review report

Kotte et al report on a very surprising adaptation problem of *E. coli*, while it is shifted from mid-exponential growth on glucose to a medium with a gluconeogenic carbon source (e.g. acetate). The result is that a significant fraction of the cell population remains dormant, those cells do not resume growth on the new carbon source, and become resistant to antibiotics that attack growing cells and less so to antibiotics that kill non-growing cells. The other cells do commence growth on acetate. The authors show that the two subpopulations emerge after the switch, due to a bistability occurring during the transition from the glycolytic to the gluconeogenic metabolic activity. Only cells that have a high enough PEP concentration induce gluconeogenesis by relieving the FBP-

mediated inhibition of Cra - the main transcription factor responsible for gluconeogenesis induction. This occurs via an unexpected positive-loop mechanism: strong feedforward activation of fructose-bisphosphatase (Fbp) by PEP causes an enhanced activity of Fbp even though the concentration of FBP - a substrate of Fbp - decreases, this relieves the inhibition of Cra and induces transcription of gluconeogenic enzymes. (In addition; lower FBP relieves production inhibition of its producing reaction and indirectly stimulates its synthesis.) Then more PEP is made, FBP reduces more, more transcription, etc: a positive loop. Cells in which FBP does not drop - or not enough - are not able to make the switch. Hence, two subpopulations of isogenic cells result by a non-genetic mechanism.

The author show convincingly that:

1. the emergence of the subpopulations has a non-genetic origin
2. the uptake rate (transporter-mediated fumarate or passive for acetate) partially determines the fraction of growing cells
3. that the anti-correlation between PEP and FBP occurs - albeit with a time course that contains gap
4. gluconeogenesis is not active on glucose growth but that mixed growth does occur
5. a model can describe the bistability phenomenon

Questions:

1. As far as I can tell, all the shift experiments in batch are done with cells still exposed to excess glucose and, therefore, growing (mid-)exponentially. The glucose limitation experiments (Figure 4B) indicate that the fraction of growing cells increases if the concentrations of glucose decrease (i.e. this happens in the chemostat when the growth rate goes down). Extrapolation of the dilution rate to zero - extrapolation to zero glucose - suggests that then α becomes close to 1. So, it appears that the following key experiment is missing (or I missed it): "What happens when in batch, glucose is depleted and the cells start growing on the acetate they have produced? What is the value of α then? I expect it to be around 1. What is then the adaptation time, is it still long? If α is close to 1 then the long lag phase is no longer due to the outgrowth of a small fraction of cells but due to some other mechanism."

Why did the authors decide to do the experiment as they have done it? In addition, the washing step as part of the shift protocol could perturb the cells to such an extent that this causes dormancy of a fraction of the population; I am missing the control experiment to rule out this possibility.

2. The fact that higher acetate values enhance α also suggests that the intrinsic property of E coli to excrete acetate to some level helps it to shift to gluconeogenesis when glucose runs out. Hence, E coli increases α itself as time progresses. How do the acetate concentrations in batch (when glucose has run out) compare to the concentration of acetate used in this paper? The interesting thing is that this phenomenon may now depend on the initial glucose amount added to the batch fermentor (or shake flask) as this determines the amount of acetate when glucose runs out. Have the authors considered doing such experiments? If not, then why not?

3. The experiments suggest that the dormant cells do not start to grow on acetate but remain dormant. This suggests that the type of bistability is actually irreversible; cells that are dormant on acetate remain dormant. This then rules out stochasticity-induced switching, but this is nowhere experimentally addressed or discussed at length. So, do the dormant cells - for instance, after antibiotics treatment on acetate to kill the growing - really not start growing spontaneously on acetate if you wait for several days? (And I mean, not due to mutations.)

4. The authors state in the abstract that the phenomenon they find is due to stochastically generated phenotypic subpopulations but they nowhere speculate about what kind of stochastic phenomena underlie this behaviour. Additional experiments using fluorescent reported studies on the enzymes controlling the ratio of PEP/FBP prior to the shift would be expected to show correlations with the probability of a cell to become dormant. Would the authors consider such experiments not required when they state that the phenomenon is stochastic and that the origin lies somehow in an imbalance between [PEP], [FBP] and FBP turnover? The least the authors should do is to speculate about the underlying origins of stochasticity in the discussion; as this may be one of the important lessons of this paper: that metabolism while operating in a deterministic regime - according to most of us - can actually display stochastic adaptive behaviour. This is of course possible when metabolism operates close to a saddle node bifurcation; then small fluctuations can cause qualitative changes. Something

like this should be raised by the authors in the discussion.

5. the authors do not show that the lag phase depends on alpha, which is should. How much longer is the lag phase as function of alpha given a fixed growth rate difference between growing and dormant cells? Is this significant when alpha is 0.5? I think not, for alpha 0.5 within one doubling time the growing population has reached its old value so the delay is negligible. And with alpha equals 0.25 within two generations. So the delay is likely not explained by the alpha value? The authors should discuss this properly.

6. The authors claim in the discussion that they explain the lag phase duration but they do not. They nowhere show the real lag phase in the natural batch situation where E coli first consumes glucose and then grows on its excreted acetate.
So how should I interpret those claims?

7. I really wonder whether this phenomenon has anything to do with bet hedging strategy as suggested by the authors in the discussion. I find this highly speculative. Partially, because the phenomenon is absent when the reverse shift is made. The phenomenon is likely due to some quirk in glycolysis control - perhaps due to some surprising kinetic problem - and it is likely not the outcome of a selective process having lead to bet-hedging behaviour. Especially not when the alpha's are much higher than a few percent; this simply amounts to biomass spoilage. So, this brings us back to the earlier question: what is a realistic alpha value, considering realistic acetate concentrations after glucose depletion during batch growth?

Minor points:

1. the authors introduce a complex concept in the second sentence of the abstract, which I would recommend to remove: "with certain bistable substrate-uptake pathways". Being familiar with bistability, fluctuations, and metabolism even I do not know what this means.

2. the second before last sentence in the abstract suggest that this mechanism "selects" cells; this sounds a bit too teleological for my taste nor do I think that the mechanism presented by the authors is an evolved trait rather a by-product of something else. I do not buy the bet-hedging hypothesis before I see an experimental illustration.

3. In the introduction the authors should describe the best studied stochasticity-induced nutrient system: the glucose-lactose diauxic shift in E coli as studied by Choi, et al, Science, 2008 and more recently by Boulineau, et al Plos One, 2013. It is known that lag-phase can be determined by stochastic non-genetic mechanisms. Other systems such as the galactose switch in yeast are also bistable and this is not mentioned either (see papers by van Oudenaarden, Bolouri, and I believe also Serrano).

4. On page 5, in the first paragraph, the sentence before last has a very complicated final part, which does not read correctly and is very hard to understand: "... and describing the exponential growth at multiple time points." Please change.

5. On page 5, the 5th sentence from the bottom speaks of 0.1% whereas the last sentence of the previous paragraph speaks of 0.01%, both in relation to 2 g/L fumarate. As far as I can tell this is a typo?

5. On page 6, the 8th sentence from the bottom needs a space between "...2009)also..."

6. In the first paragraph of the starting section on page 7 the authors argue that the Monod type of relation indicates that the rate of acetate uptake is limiting growth. This is not true, as intracellular processes can be limiting equally well. This would only be true if the Monod constant equals the Km of the transporter and when the transporter has a growth rate control coefficient of one 1 which we do know without an experiment.

7. On page 8, in the fifth sentence from the bottom the authors speak of "merged flux" I do not understand what this means, please rephrase.

8. On page 12, the sentence starting with "Taken together," should be split into; it is too long and does not run properly

Editor comments

The reviewers find the main observations interesting and novel. Without repeating all the points listed in their reports, the reviewers raise concerns with regard to three aspects of the study:

- The nature of the stochastic process that underlies the generation of the two phenotypic sub-populations. While identifying the exact source of stochasticity might be difficult (even though we note that reviewer #2 provides constructive suggestions), a better discussion of this point is required.

We have added a better discussion of this point.

- The relationship between the fraction of growing cells (α) and the length of the resulting (apparent) lag phase should be clarified (see points raised by reviewer #2).

Yes, this connection was indeed not made very clear in the last version; but it is clarified with an additional figure in the Supplement.

- The most substantial concerns refer to the proposed mechanisms that would explain the observed responsive phenotypic bistability--in particular the increase in gluconeogenic flux but drop in FBP levels that creates the double negative feedback loop via Cra. While measuring ALL enzymatic activities might be challenging, more direct experimental evidence for the changes in glucogenogenic fluxes and for the impact of PEP on Fbp activity seems to be important, as noted by reviewer #1.

We did not measure the enzymatic activities, but in response to this reviewer's request to prove "conclusively which gluconeogenic enzymes are being up-regulated and to what extent" (cf. comment 4), we measured the abundances of these enzymes and report them below.

Reviewer #1:

SUMMARY

The authors have written a paper describing a very intriguing and comprehensive study which demonstrates phenotypic bistability in Escherichia coli central carbon metabolism in response to a shift from glycolytic to gluconeogenic growth, and elucidates the nature of this phenomenon. The study explored how isogenic populations of bacterial cells are able to produce alternative phenotypes through stochastic means, conferring upon the population a degree of adaptability to environmental fluctuations. The study presents a diverse array of experiments which characterise the stochastic growth responses of these cells including: fluorescence staining, flow cytometry, LC/MS, and a host of studies with mutants and cells containing inducibly expressed genes and reporter plasmids.

The primary conclusions of this study were that the lag in growth observed in E. coli in response to a shift from glycolytic to gluconeogenic substrates can be attributed to a responsive stochastic diversification of an isogenic population of cells with a shared phenotype into two sub-populations with distinct and stable growth rates. Additionally, it was shown that the proportion of cells switching to the high growth rate phenotype was dependent upon gluconeogenic substrate concentration, and by implication, on gluconeogenic flux and the activity of the global transcription factor Cra. A mechanism was proposed for the bistable diversification of this system involving Cra and fructose 1,6-bisphosphate (FBP), and explored by the construction and scrutinizing of a simplified core model. On this basis, the authors speculate that it is indeed possible for the observed bistability to be achieved by a feed forward effect between a flux-dependent gluconeogenic metabolite concentration and its allosteric activation of the FBP consuming fructose 1,6-bisphosphatase (FBPase) reaction, which results in the the alleviation of the inhibition that FBP exerts on Cra.

GENERAL REMARKS

The authors present a well-rounded experimental study involving a varied array of evidence. The conclusions drawn from the experimental evidence appear to be valid. This paper further elucidates the nuanced mechanisms involved in the regulation of glycolytic and gluconeogenic metabolism, and this discovery has implications for, amongst others, industry, where the presence of growing and non-growing phenotypes are an important concern as E. coli is a favoured protein expression vector; and for medicine, as the discussion about antibiotic sensitivity demonstrated.

The intention of the authors to provide a putative mechanistic explanation for the observed experimental behaviour by constructing a mathematical model of the system is praiseworthy. However, the constructed model was oversimplified and in our opinion the conclusions drawn with regard to the proposed bistability mechanism are not fully justified.

The purpose of the model is to explain the multitude of experimental observations, i.e. to bring these into context. Towards this purpose, the model has the right complexity, as it is the simplest mechanistic model we could find that agrees with *all* experimental observations. Hence, in our view, any additional complexity is not only unnecessary, but even “harmful”. With increasing complexity, the model would increasingly lose its focus on the core, flux-sensor-centric mechanism that generates bistability.

This reviewer suggests to add substantial complexity to the model (see also comments below). The purpose of this added complexity seems to be an investigation of the influence of factors outside of the current model's system boundary onto the modeled mechanism. However, due to inherent uncertainty in modeling, in our view, even a very sophisticated mechanistic model is not the correct tool to research the influence of these external factors when the living cell is readily available for investigation. To make this point exceedingly clear, we do not agree with this reviewer on the purpose towards which the model is geared: We intend to use the simplest meaningful model as a tool to understand what ties our experimental observations together. We believe the model does an excellent job in this regard and would very much like to keep it as it is; we do not intend to use the

model as a tool to foray into areas that we cannot back up with experimental data or even to use the model as a substitute for experiments.

MAJOR CRITICISMS

The core model developed in the latter sections of the paper is too simplistic and we found the conclusions drawn from this model to be too speculative. Some weaknesses of the model are:

1. It is not necessarily true that the gluconeogenic substrate X maintains a simple proportional relationship to flux. Correlation does not imply causation, and moreover the gluconeogenic flux was not measured directly but calculated on the basis of an FBA model (Fig. 4A). The underlying mechanistic causes may well lie somewhere else. For example, a number of regulatory relationships span the central metabolic pathways in E. coli, often incorporating the energy charge and redox state of the cell; facets that were excluded from this model.

First, note: The acetate uptake flux at low acetate concentrations unfortunately cannot be directly measured and thus had to be inferred via a model; see the reasons outlined at the end of this comment. However, for the experiment, where we modulated the fumarate transporter abundance, we have directly measured uptake fluxes (cf. Supplementary Figure 6) and could show that these influence α .

In Fig. 4A we observe a correlation between flux and Cra activity. Mechanistically, this correlation is eventually established in the following manner: flux generates a certain FBP level, which in turn modulates Cra activity. However, many other factors could interfere or could be involved in this chain of causalities. What we understand from this reviewer is that he/she would like us to resolve and include all the possible additional biochemical complexities. We think that (i) it is basically impossible to correctly model all or even most of other influencing factors and that (ii) it would not contribute to our story. Please try to envision what would be necessary to mathematically model the biochemical system (in a mechanistically comprehensive manner) which is responsible to establish the observed relationship between flux and Cra activities (transcript levels, RNA polymerase concentrations, protein levels, TF abundances, metabolite levels, growth rate and general physiology, etc.). Even if one would go through such a major effort, such a model would very likely be still incomplete (and probably still far from the true biological reality). Ultimately, even if we would have such a model, what kind of evidence would we gain from it?

Our simple model, which is consistent with all our experimental observations but does not describe the complete underlying biochemistry, serves only two purposes. First, it is an explanatory model making the understanding of the bistable mechanism easier for the reader. Second, we use the model to test if once we bring together the experimental observations that we made, the system could indeed generate flux-dependent bistability. We use this as a confirmation whether the experimentally derived hypothesis about the bistability-generating mechanism is sufficient and whether the system could actually behave as we propose.

Explanation why determining uptake rate at the different acetate concentrations is inherently not feasible:

In batch cultures, uptake rates are usually determined by measuring the change in extracellular substrate concentration and biomass over time and by fitting this data to a model describing the cellular growth and the growth-dependent substrate depletion. Key elements for this data fitting are that (i) changes in extracellular substrate concentrations are measurable and (ii) that at the same time cells grow at a constant growth rate. Steady-state growth is only ensured if the changing substrate concentration (which cannot be avoided during this experiment) is in turn not affecting the substrate uptake rate and thus growth rate. However, it is not the case here, as acetate concentrations that allow *E.coli* growth are in the same range as the K_m value for this substrate. Thus, we had to indirectly estimate the acetate uptake rates from the growth rate data.

2. *It is true that with reduced activities of the enzymes catalysing the reactions from PEP to FBP, in a simple pathway a higher concentration of PEP would be required to achieve the same flux (p. 35, Supplementary Information). However, the extent of this required increase in PEP depends on how far the reactions are from equilibrium, and if there is sufficient excess capacity, the same steady-state flux could in principle be maintained by a virtually unaltered PEP concentration even if enzyme levels are decreased substantially. In general, the control of the steady state concentration of any metabolite in a network is shared amongst the members of the network, and close-to-equilibrium reactions tend to have low control coefficients.*

In the supplementary text, which is the basis of this reviewer's comment, we explain the consequences of the Cra-regulated repression of enzymes between the metabolites PEP and FBP. We argued that if this Cra-mediated repression has an effect on the functioning of the proposed bistability-generating mechanisms, it could only be that the concentration of PEP increases – making its feedforward activation even stronger. However, this aspect, discussed and illustrated in the supplement, is only a side aspect. What is essential, is that we confirmed experimentally that PEP concentrations are higher at higher extracellular acetate concentration (and thus higher intracellular flux). Our model captures, in a coarse-grained manner, what we observe experimentally. As PEP is a highly connected metabolite, we do not claim that we have, in the model, implemented the exact cause for these increasing PEP levels. We are however content with having established, in the model, one mechanism for increasing PEP levels with flux. Of course, other reactions involving PEP and other unpredicted interactions may contribute – but for our purpose, this is not important.

The regulation of the enzymes between PEP and FBP, which is not part of the model, is not crucial for the generation of bistability according to the proposed mechanism (as we show in the Supplement). The regulation may only modulate, but not fundamentally change the bifurcation properties. If it does modulate the mechanism, it is capable only of enhancing the bistable region.

3. *Since a detailed kinetic model of E. coli central carbon metabolism has been published recently (Peskov et al. 2012, FEBS Journal), it should be possible to produce a quantitative model of all the molecular interactions involved in central carbon metabolism to identify the actual underlying molecular mechanisms. Since PEP exists at a complex regulatory node in central carbon metabolism, its treatment in the current model is too superficial.*

The mechanism responsible for the generation of bistability includes a feedback from metabolism (via the metabolite FBP) to the modulation of Cra activity and the transcriptional layer back to the production of enzymes for reactions required during growth on acetate. The detailed model of Peskov et al. does not contain the any transcriptional regulation, which however is a crucial element in the uncovered mechanisms. Part of the bistability-generating mechanism is therefore outside of the system boundary of the Peskov model and hence the Peskov model is not suited for investigations here.

In addition to that, please remember that we have *experimentally* verified the negative correlation of the metabolite FBP with gluconeogenic flux. It is this experimentally observed negative flux dependency of FBP levels that is crucial and decisive for the proposed mechanism. Which metabolic regulations create this inverse flux dependency of FBP levels is interesting, however, is not of importance for our work. In the model, we have included the feed-forward activation of Fbp by PEP as a probable enzymatic regulation to achieve the inverse flux dependency. We have even performed experiments to demonstrate a positive correlation of PEP levels with flux. Even if another metabolic regulation mechanism was responsible for the inversion of FBP levels, it would not matter for the primary bistability-generating mechanism in which Cra levels are regulated by (experimentally verified) flux-signaling FBP levels.

4. *The underlying molecular mechanism can be further elucidated by measuring the activities of all the enzymes in the gluconeogenic pathway from acetate to FBP using classical biochemical activity assays (methods have been described) to provide a better quantitative picture of the metabolic reprogramming that occurs. This should be done at t=0 directly after the switch, and at t=15h when the majority of the culture would be made up of fast-growing cells. This would prove conclusively which gluconeogenic enzymes are being upregulated and to what extent.*

The reviewer asks us to determine enzyme activities at two different time points (at $t=0$ and at $t=15$ hours with the first time point basically representing steady-state growth on glucose and the second one steady-state on acetate). In fact, we had recently quantitatively determined abundance of 2000 *E. coli* proteins under a number of different growth conditions, amongst them glucose (representing $t=0$) and acetate. This data set, which is a part of a manuscript currently under review elsewhere, contains the requested data. Please find the concentrations of enzymes in the gluconeogenic pathway from acetate to FBP in the table below (enzymes highlighted in bold are gluconeogenic proteins with significantly up-regulated expression levels). This data shows which gluconeogenic enzymes are being up-regulated and to what extent.

	copies / fL cell volume		concentration fold change
	glucose	acetate	
fbaB	28	26	0.93
fbaA	4573	4366	0.95
tpiA	2689	3811	1.42
gapA	8600	7224	0.84
pgk	5375	6614	1.23
gpmM	87	58	0.67
pgmA	3420	3987	1.17
eno	5605	5202	0.93
ppsA	260	293	1.13
pckA	574	2057	3.59
mdh	6267	19968	3.19
maeB	160	222	1.38
maeA	332	302	0.91
aceB	147	128	0.87
aceA	4775	26791	5.61
acnA	69	223	3.24
acnB	1449	2659	1.83
gltA	1189	1799	1.51
acs	58	149	2.59
pta	386	252	0.65
ackA	952	723	0.76

5. It is not clear in the literature whether PEP acts as an activator (Hines 2006) or inhibitor (Babul 1983) of the FBPase reaction (or whether a more complex allosteric relationship exists). This should be clarified experimentally as it is fundamental to the authors' hypothesis.

In fact, this requested experimental clarification was done last year, where it was demonstrated that PEP activates FBPase *in vivo* (cf. Fig 2d in Nature Biotechnology paper, PMID 23455438). Notably, the activation of the FBPase through PEP was the third most prominent interaction in this study (out of the 90 tested). Thus, this study has already confirmed (even *in vivo*) what the reviewer requested. We have added this reference to the revised version of the manuscript (page 10).

6. In conclusion, the experimental evidence for responsive diversification into two separate subpopulations upon switch to a gluconeogenic substrate is convincing, as is the requirement for Cra and Fbp to observe this effect. However, the core model put forward is too speculative to provide convincing evidence for the hypothesised mechanism. To be convincing, the hypothesis should be tested using a detailed quantitative molecular-level model of central carbon metabolism

in E. coli. In addition, techniques such as metabolic control analysis can provide insight into the control and regulation of fluxes and metabolite concentrations under the different conditions.

As mentioned above, we feel that evidence for a proposed molecular mechanism should come from experiments and not from a modeling exercise. We feel that we have generated sufficient experimental evidence for the proposed mechanism. In our view, there is no relevant additional insight that a complex mechanistic model of central metabolism, metabolic control analysis and further investigations into the control and regulation of fluxes and metabolite concentrations could yield. Our current model is as small and focused as we could make it to explain our experimental observations and the observed bistability.

MINOR CRITICISMS

1. A number of grammatical errors were discovered in the text, and the quality of the language use in general could be improved.

A (non-exhaustive) list of examples:

- the verb "describing" in the middle of page 5 seems inappropriate

We have changed “and describing the exponential growth” to “and calculating the growth rate of”. (Page 5, 2nd paragraph from the top).

- "rationale", not "rational"

Thanks. We have made the change.

- "kinetics", not "kinetic"

Thanks. We have made the change.

2. The figure reference to Fig. 4B, D (p. 13, 6 lines up) is wrong. Figure cross-references should be re-checked.

Thank you for pointing to these errors. We corrected them and also re-checked the manuscript and supplement again for errors.

Reviewer #2:

Kotte et al report on a very surprising adaptation problem of E. coli, while it is shifted from mid-exponential growth on glucose to a medium with a gluconeogenic carbon source (e.g. acetate). The result is that a significant fraction of the cell population remains dormant, those cells does not resume growth on the new carbon source, and become resistant to antibiotics that attack growing cells and less so to antibiotics that kill non-growing cells. The other cells do commence growth on acetate. The authors show that the two subpopulations emerge after the switch, due to a bistability occurring during the transition from the glycolytic to the gluconeogenic metabolic activity. Only cells that have a high enough PEP concentration induce gluconeogenesis by relieving the FBP-mediated inhibition of Cra - the main transcription factor factor responsible for gluconeogenesis induction. This occurs via an unexpected positive-loop mechanism: strong feedforward activation of fructose-bisphosphatase (Fbp) by PEP causes an enhanced activity of Fbp even though the concentration of FBP - a substrate of Fbp - decreases, this relieves the inhibition of Cra and induces transcription of gluconeogenic enzymes. (In addition; lower FBP relieves production inhibition of its producing reaction and indirectly stimulates its synthesis.) Then more PEP is made, FBP reduces more, more transcription, etc: a positive loop. Cells in which FBP does not drop - or

not enough - are not able to make the switch. Hence, two subpopulations of isogenic cells result by a non-genetic mechanism.

Thank you very much for the very positive remarks and the excellent summary – we could not have summarized it better.

The author show convincingly that:

- 1. the emergence of the subpopulations has a non-genetic origin*
- 2. the uptake rate (transporter-mediated fumarate or passive for acetate) partially determines the fraction of growing cells*
- 3. that the anti-correlation between PEP and FBP occurs - albeit with a time course that contains gap*
- 4. gluconeogenesis is not active on glucose growth but that mixed growth does occur*
- 5. a model can describe the bistability phenomenon*

Questions:

1. As far as I can tell, all the shift experiments in batch are done with cells still exposed to excess glucose and, therefore, growing (mid-)exponentially. The glucose limitation experiments (Figure 4B) indicate that the fraction of growing cells increases if the concentrations of glucose decrease (i.e. this happens in the chemostat when the growth rate goes down). Extrapolation of the dilution rate to zero - extrapolation to zero glucose - suggests that then alpha becomes close to 1. So, it appears that the following key experiment is missing (or I missed it): "What happens when in batch, glucose is depleted and the cells start growing on the acetate they have produced? What is the value of alpha then? I expect it to be around 1. What is then the adaptation time, is it still long? If alpha is close to 1 then the long lag phase is no longer due to the outgrowth of a small fraction of cells but due to some other mechanism."

Why did the authors decide to do the experiment as they have done it? In addition, the washing step as part of the shift protocol could perturb the cells to such an extent that this causes dormancy of a fraction of the population; I am missing the control experiment to rule out this possibility.

The reviewer wonders whether the alpha value would be 1 if one would extrapolate to zero glucose concentration or "zero" growth rate on glucose. First, note that in *E. coli* glucose limited chemostat cultures the concentration of glucose is basically always zero, independently on the dilution rate (because of *E. coli*'s extremely low affinity constant for glucose). So, the glucose concentration can be excluded as an effector for alpha. The steady-state growth rate on glucose, however, has an effect, which is what we show in Figure 4B. Different growth rates on glucose are accompanied by different intracellular fluxes, which are being "measured" by *E. coli* (Kochanowski et al, PNAS, 2013) and used for flux-dependent regulation of gene expression. At low glycolytic fluxes (=at low chemostat dilution rates) the gene/protein expression profile has been found to be more shifted to metabolism of alternative carbon sources/TCA cycle/etc. Such a protein expression profile is beneficial for making the transition to a gluconeogenic carbon source, which manifests as a high alpha value (which is what we found).

Therefore, the actual answer to the question of the reviewer whether the alpha would be 1 in a real diauxic shift experiment depends on a number of factors: (i) How fast does the glucose run out, because if the glucose concentration range around the very low affinity constant is passed very quickly then there would be not enough time for the cell to adjust its gene/protein expression accordingly. (ii) Would the gene/protein expression profile at the lowest possible growth rate on

glucose be such that after the switch to the gluconeogenic carbon source in all cells a high enough gluconeogenic flux can be achieved that would allow them to assume the growing state?

We do not find it highly relevant to investigate into what the alpha would be in such a glucose-depletion experiment, because – as said above – this will likely also depend on how fast the glucose runs out and this will in turn depend on the number of cells present at that time etc. Further, we are not so much interested in the “true” alpha value (because: what would be the true natural shift condition?), but rather in how we can influence the alpha. Only this tell us something about the underlying mechanism. To identify this mechanism, we did the shift experiments in a way that is may be a bit artificial (washing and staining), but these controlled artificial shift experiments allowed us to identify the underlying mechanism.

Note, even different *E. coli* wild-type strains show largely different alpha values; ranging from about 1×10^{-4} (switch from 5g/L glucose to 0.5 g/L acetate, our unpublished results, strain SS205 received from M. Doebeli, mentioned in PMID: 15068343) to the values we report in our paper. So, investigating about the true alpha in a “true natural shift” (whatever this might be), is in at least our view not the core point of our work.

In fact, only by going through the stain/washing procedure, we could unambiguously determine that there are growing and non-growing cells present, because the assay does not depend on debatable fluorescent or other reporters. Likely, the washing has an effect on the alpha. However, because in all our experiments the cells go through the same washing/staining procedure and because we can influence the alpha by different means before and after the switch, we could unravel the underlying mechanism.

Still, we have performed an experiment with a real diauxic shift to add evidence that what we concluded from the “artificial shifts” (i.e. that acetate concentration influences alpha and therefore the lag phase) would also hold in this case. In this experiment, we have reduced the amount of glucose present in the medium to 0.25 g/L and supplemented it with 0.5 g/L or 0.75 g/L acetate (already at the start of the culture). We had to do this experiment with low glucose concentrations, because otherwise the cell concentration would have been too high at the moment of glucose depletion and we would not have been able to observe a biomass increase on (the low concentrations of) acetate. Therefore, acetate had to be spiked so we had different acetate concentrations in different replicates (but still corresponding to acetate concentrations at the moment when glucose is depleted in a normal batch culture in both conditions). Of course, here we could not stain the cells but could only monitor the duration of the lag phase. Consistent with the concentration-dependency that we found using the artificial switches, the apparent lag period is shorter for the culture in which 0.75 g/L acetate was added than in the culture in which 0.5 g/L acetate was added. This fact, consistent with all our results, while not sufficient to prove emergence of two populations, is at least necessary for the emergence of two populations and their ratios being different in both conditions.

Growth curves of *E.coli* cultures during glucose-acetate diauxic shifts. Open squares – cultures with 0.75 g L^{-1} acetate added, open circles – cultures with 0.5 g L^{-1} acetate added. Grey area indicates growth period with no glucose present in the medium. Due to acetate production, the actual acetate concentrations at the moment of glucose depletion were $0.53 \pm 0.03 \text{ g L}^{-1}$ in the culture with 0.5 g L^{-1} acetate added and $0.81 \pm 0.01 \text{ g L}^{-1}$ in the culture with 0.75 g L^{-1} acetate added. Glucose and acetate concentrations were determined by HPLC-RID analysis. OD normalized to 1 at the moment of glucose depletion. Average data from triplicate experiments shown. Error bars indicate one standard deviation.

2. The fact that higher acetate values enhance alpha also suggests that the intrinsic property of E coli to excrete acetate to some level helps it to shift to gluconeogenesis when glucose runs out. Hence, E coli increases alpha itself as time progresses. How do the acetate concentrations in batch (when glucose has run out) compare to the concentration of acetate used in this paper? The interesting thing is that this phenomenon may now depend on the initial glucose amount added to the batch fermentor (or shake flask) as this determines the amount of acetate when glucose runs out. Have the authors considered doing such experiments? If not, then why not?

Yes, we agree with the reviewer: at higher glucose concentrations, more acetate would be produced which should lead to more cells switching to growth on acetate and therefore, expectedly, a shortened lag phase. However, higher initial glucose concentrations result in more acetate produced, which also leads to differently strong drops in pH. This influences the growth rate on glucose, which in turn additionally affects the alpha. Thus, such - in principle - simple experiments would unfortunately not lead to conclusive results. For this reason, we performed the experiment shown in response to the previous comment, where we actually mimicked what this reviewer wondered in his comment. In fact, we saw that higher acetate concentrations indeed lead to shortened lag phases.

With regard to the other question: The acetate concentrations we used in our experiments are comparable to the acetate concentrations produced by *E. coli* growing in typical lab batch experiments on glucose (e.g. 5 g/L), at the moment of glucose depletion. Reported acetate yields range from 0.07 to $0.184 \text{ [g acetate / g glucose]}$ (<http://aem.asm.org/content/56/4/1004.full.pdf>). We have added a statement to the manuscript that the acetate concentrations that we used are in the same range as the acetate concentrations occurring in normal lab glucose cultures and added the appropriate reference (page 24, Figure 3 caption).

3. The experiments suggest that the dormant cells do not start to grow on acetate but remain dormant. This suggests that the type of bistability is actually irreversible; cells that are dormant on acetate remain dormant. This then rules out stochasticity-induced switching, but this is nowhere experimentally addressed or discussed at length. So, do the dormant cells - for instance, after antibiotics treatment on acetate to kill the growing - really not start growing spontaneously on acetate if you wait for several days? (And I mean, not due to mutations.)

We feel that the reviewer might have overlooked this point in the previous manuscript. We mentioned this point in the main text and referred the reader to an experiment shown in the Supplementary figure 4C. This figure shows that the number of non-growing cells remains constant and that the non-growing cells retain their fluorescence, indicating that there is no stochastic switching between the two states.

Switching cells to fumarate is the most favorable switch to observe eventual stochastic switching, as this is the condition where alpha is lowest, allowing for the longest observation. (Note that the observation period is limited by the number of growing cells that at some point outnumber the non-growing cells.) Using the switch to fumarate, we can safely state that at least until 31 h after the switch the non-growing cells really do not start growing.

Unfortunately, we cannot test for longer periods, because using antibiotics as suggested by this reviewer brings the problem of having to remove the antibiotic, which would involve perturbation of the cells through washing. Additionally, antibiotic treatment will likely lead to cell lysis of the growing cells, which could provide the non-growing cells with nutrients that they could use to wake up (because of new nutrient availability) and not for stochastic reasons.

4. *The authors state in the abstract that the phenomenon they find is due to stochastically generated phenotypic subpopulations but they nowhere speculate about what kind of stochastic phenomena underlie this behaviour. Additional experiments using fluorescent reported studies on the enzymes controlling the ratio of PEP/FBP prior to the shift would be expected to show correlations with the probability of a cell to become dormant. Would the authors consider such experiments not required when they state that the phenomenon is stochastic and that the origin lies somehow in an imbalance between [PEP], [FBP] and FBP turnover? The least the authors should do is to speculate about the underlying origins of stochasticity in the discussion; as this may be one of the important lessons of this paper: that metabolism while operating in a deterministic regime - according to most of us - can actually display stochastic adaptive behaviour. This is of course possible when metabolism operates close to a saddle node bifurcation; then small fluctuations can cause qualitative changes. Something like this should be raised by the authors in the discussion.*

We agree with the reviewer and added a short discussion on the origins of the stochasticity (page 11). Overall, we have determined that *varying flux to FBP* is the decisive factor for whether a cell will go for the growing or non-growing phenotype. This stochastic variation in flux could have multitude of different underlying reasons. In fact, every enzyme from the substrate transporter all the way up to FBP could be a flux bottleneck and even in different cells the flux bottlenecks could be at different enzymes. Furthermore, it could be that the abundance of Cra is variable between cells. Overall, there could be many different reasons and these reasons could even be different between cells, why certain cells can accomplish a sufficient high enough flux and others cannot.

5. *the authors do not show that the lag phase depends on alpha, which is should. How much longer is the lag phase as function of alpha given a fixed growth rate difference between growing and dormant cells? Is this significant when alpha is 0.5? I think not, for alpha 0.5 within one doubling time the growing population has reached its old value so the delay is negligible. And with alpha equals 0.25 within two generations. So the delay is likely not explained by the alpha value? The authors should discuss this properly.*

Indeed, with acetate the effects of alpha on the lag phase are not super obvious and indeed there is always still also a normal “lag phase”, i.e. a time where growing-cells speed up to reach the growth rate on the new carbon source.

Instead of showing the how alpha “translates” into lag phases, which is indeed a point that we did not show in the last manuscript, we added data from our experiments where we switched cells from the glucose chemostat cultures to a batch culture containing 2g/L of fumarate. Here, the reader can now clearly see the relationship between alpha and the lag phase duration (new Supplementary Figure 7).

6. *The authors claim in the discussion that they explain the lag phase duration but they do not. They nowhere show the real lag phase in the natural batch situation where E coli first consumes glucose and then grows on its excreted acetate. So how should I interpret those claims?*

The reviewer is right; we indeed did not illustrate that non-growing cells (“alpha”) are a cause for observed apparent lag phases. We now do this with the newly added Supplementary Figure 7 (cf. response to comment 5). As for the “real lag phase in natural batch situation”, we would like to refer the reviewer to our response to comment 1.

7. *I really wonder whether this phenomenon has anything to do with bet hedging strategy as suggested by the authors in the discussion. I find this highly speculative. Partially, because the phenomenon is absent when the reverse shift is made. The phenomenon is likely due to some quirk in glycolysis control - perhaps due to some surprising kinetic problem - and it is likely not the outcome of a selective process having lead to bet-hedging behaviour. Especially not when the alpha's are much higher than a few percent; this simply amounts to biomass spoilage. So, this brings us back to the earlier question: what is a realistic alpha value, considering realistic acetate concentrations after glucose depletion during batch growth?*

First, again, coming back to our answer to comment 1, we think that one cannot state what a “natural” shift would be for *E. coli*. Even the “normal” diauxic shift that we observe in lab might not be something that *E. coli* encounters in its true environment. Further, we feel that one should not place too much emphasis on the actual alpha values, because we found that there are even strong differences between different *E. coli* wild-type lab strains (see also respond to comment 1).

Bet-hedging, as used in the microbiological literature, is mostly associated with the occurrence of distinctively different phenotypes in one condition, where one of them has lower fitness, but then takes over after an environmental change (and vice versa). While in our case we do not have *distinctively* different phenotypes (on glucose), we have *gradually* different phenotypes on glucose. In particular, we have cells growing with different rates on glucose. Those that grow fast have a protein expression profile that is less beneficial when the condition changes to a gluconeogenic carbon source and vice versa. Thus, cells on glucose have a gradually different fitness (i.e. they have different growth rates) and the ones with lower fitness (slower growth) have an advantage if the environment changes to a gluconeogenic carbon source. What we do is just extending the concept of bet-hedging as used in microbiology from the typically considered case with *distinctively different phenotypes* to a *gradually different phenotype*, with the existence of a trade-off intact (i.e. cells being more fit on one condition means being less fit after an environmental change). Because the environmental conditions before and after the change affect the phenotype distributions, which is a property of conditional optimal strategy, we classify the phenomenon described by us as “conditional bet-hedging”, as suggested in (<http://dx.doi.org/10.1002/bies.201000127>). We have added this reference (Page 14, last paragraph).

We have extensively discussed the question whether what we see here can be considered as bet-hedging with our colleagues Oscar Kuipers and Jan-Willem Veening (who work at the same university as we do). They have extensively used the term in the past to describe similar behaviors and who have even written a review that describes how to classify different evolutionary stable strategies, including bet-hedging. They agreed that what we see could be described as bet-hedging. Because, of course, we cannot deliver a formal proof, we added the term “eventually” to the respective sentence, alerting the reader that we do not provide the ultimate proof.

Minor points:

1. the authors introduce a complex concept in the second sentence of the abstract, which I would recommend to remove: "with certain bistable substrate-uptake pathways". Being familiar with bistability, fluctuations, and metabolism even I do not know what this means.

Here, we intended to refer to exactly the papers mentioned under the point 3 below, i.e. studies which found bistability in the expression of certain uptake pathway. But in fact calling it “bistable pathways” indeed doesn’t make much sense. We changed it to “bistably expressed substrate-uptake pathways“ (page 2).

2. the second before last sentence in the abstract suggest that this mechanism "selects" cells; this sounds a bit too teleological for my taste nor do I think that the mechanism presented by the authors is an evolved trait rather a by-product of something else. I do not buy the bet-hedging hypothesis before I see an experimental illustration.

We changed it to “and only allows the growth of cells achieving sufficiently high gluconeogenic flux”. (page 2).

3. In the introduction the authors should describe the best studied stochasticity-induced nutrient system: the glucose-lactose diauxic shift in E coli as studied by Choi, et al, Science, 2008 and more recently by Boulineau, et al Plos One, 2013. It is known that lag-phase can be determined by stochastic non-genetic mechanisms. Other systems such as the galactose switch in yeast are also bistable and this is not mentioned either (see papers by van Oudenaarden, Bolouri, and I believe also Serrano).

In order to keep the introduction focused, we would like to refrain from including papers that studied graded response to nutrient shifts. We only included the papers that reported a bistable response, such as in the papers from van Oudenaarden and Choi. The Serrano papers only covered artificial (engineered) circuits.

4. On page 5, in the first paragraph, the sentence before last has a very complicated final part, which does not read correctly and is very hard to understand: "... and describing the exponential growth at multiple time points." Please change.

Reviewer 1 had the same point and we have corrected that part of the manuscript to make it clearer (page 5).

5. On page 5, the 5th sentence from the bottom speaks of 0.1% whereas the last sentence of the previous paragraph speaks of 0.01%, both in relation to 2 g/L fumarate. As far as I can tell this is a typo?

No, this is not a mistake. We indeed speak of 0.1%, but the other value that we mentioned was $\alpha = 0.001$ (a fraction and not percent).

5. On page 6, the 8th sentence from the bottom needs a space between "...2009)also..."

Thank you for pointing it out. We corrected it.

6. In the first paragraph of the starting section on page 7 the authors argue that the Monod type of relation indicates that the rate of acetate uptake is limiting growth. This is not true, as intracellular processes can be limiting equally well. This would only be true if the Monod constant equals the K_m of the transporter and when the transporter has a growth rate control coefficient of one 1 which we do know without an experiment.

The reviewer is right. What we should have said (and what we now say) is that “the acetate-concentration-dependency of the growth rate that we observe suggests that the extracellular acetate concentration apparently influences the acetate influx and the flux into central metabolism” (page 7). Attributing this observation to a specific limitation anywhere (i.e. the uptake) is not necessary for our story, although the agreement between our observed “Monod constant” (0.5 g/L, cf. Fig 3A) and the K_m value reported for the acetate consuming enzyme Ack (0.42 g/L) is striking. Nevertheless, because it is not important for our story to speculate WHERE this acetate-concentration dependent limitation resides, it is important to know that the acetate-concentration apparently limits flux. This is what we now say. We would like to thank the reviewer to drawing our attention to this “sloppiness”.

7. On page 8, in the fifth sentence from the bottom the authors speak of "merged flux" I do not understand what this means, please rephrase.

Here, we meant that gluconeogenic carbon sources entering the TCA cycle all merge at (...). We rephrased this sentence to make it clearer (page 8).

8. On page 12, the sentence starting with "Taken together," should be split into; it is too long and does not run properly

Thank you for pointing this out. We have made the necessary change (page 11).

Thank you again for submitting your revised work to Molecular Systems Biology. We have now heard back from the referee who accepted to evaluate your revision. As you will see, this reviewer is now fully supportive and I am please to inform you that we will be able to accept your manuscript for publication in Molecular Systems Biology pending the following minor amendments:

- We would kindly ask you to briefly discuss the works of Solopova et al 2014 (Bet-hedging during bacterial diauxic shift) and van Heerden et al 2014 (Lost in transition: start-up of glycolysis yields subpopulations of nongrowing cells).
- For figures that show key quantitative data, we would strongly encourage you to provide the associated source data files (see also <http://msb.embopress.org/authorguide#a3.4.3>).
- Please provide your ODE model in a machine-readable format (SBML in principle). We would also strongly recommend to deposit the model to BioModels and include the relevant accession number.
- Please remove the long nucleotide sequences from the Supplementary Information file and provide them as separate text files (FASTA).

Reviewer #2:

I went through the replies of the authors to my comments. In my opinion, the authors have dealt adequately with all comments. I recommend acceptance of the current version of the manuscript.

As requested, we removed the primer sequences from the supplement and provided a text file in FASTA format.

We also uploaded an Excel file with the data of most figures. Because of the manuscript's terribly long history (14 man-years of work, with several people involved in different countries), it was more efficient for us to digitize the data, rather than digging into old file archives of different people. I hope that this is ok.

We added the two references that you mentioned to the discussion section.

As for providing the model in SMBL format: The model that we built (in Matlab) is just an analysis model, i.e. its value lies in the analyses done with it (as described in the supplement) and not in the model itself. It is so small that any modeler could set it up in very short time. In fact, we have now developed one in SMBL, and look how small it is:

Still, I would prefer to not submit the model. This model is so small that everybody can do it within little time him or herself. If we give this model as an SMBL file to the community, people will eventually simulate it (maybe without thinking too much about it). The actual system behavior that one obtains with this model, however, completely depends on the parameter values. I fear that researchers who quickly download models and run them would not consider this. Instead, researchers who really want to analyze and understand this system will build this model themselves without much effort on the basis of the equations that we provide in the supplement.

Although I absolutely support MSB's strive to provide data and models to the community, here, because the model is so small and its value really lies in the analysis of it, I feel that just providing it would not help anybody, but researchers anyhow face the constant struggle to deal with the ever increasing information avalanche.

I sincerely hope you agree... It is not that we are refraining from preparing the SMBL model. We actually took the time to develop an SBML version, but because I feel we do more harm than good, in this case it would be better not to include it.